


# The Effects of Sea Spray and Atmosphere-Wave Coupling on Air-Sea Exchange during Tropical Cyclone

Nikhil Garg[1], Ng Yin Kwee Eddie[2], and Srikanth Narasimalu[3]

[1,2]School of Mechanical and Aerospace Engineering, Nanyang Technological University, Singapore
[3]Energy Research Institute (ERI@N), Nanyang Technological University, Singapore

*Correspondence to:* Nikhil Garg (nikhil003@e.ntu.edu.sg)

**Abstract.** The study investigates the role of air-sea interface using numerical simulations of an Atlantic Hurricane Arthur (2014). More specifically, present study aims to discern the role ocean surface waves and sea spray play in modulating the intensity and structure of a tropical cyclone (TC). To investigate the effects of ocean surface waves and sea spray, numerical simulations were carried out using a coupled atmosphere-wave model where a sea spray microphysical model was incorporated within the coupled model. Furthermore, this study also explores how sea spray generation can be modelled using wave energy dissipation due to whitecaps, where whitecaps are considered as the primary mode of spray droplets generation at hurricane intensity wind speeds. Three different numerical simulations including sea state dependent momentum flux, sea spray mediated heat flux and combination of former two processes with sea spray mediated momentum flux were conducted. The foregoing numerical simulations were evaluated against the National Data Buoy Center (NDBC) buoy and satellite altimeter measurements as well as a control simulation using an uncoupled atmosphere model. The results indicate that the model simulations were able to capture the storm track and intensity, where the surface wave coupling results in a stronger TC. Moreover, it is also noted that when only spray mediated heat fluxes are applied in conjunction with sea state dependent momentum flux, they result in a slightly weaker TC, albeit stronger compared to the control simulation. However, when spray mediated momentum flux is applied together with spray heat fluxes, it results in a comparably stronger TC. The results presented here alludes to the role surface friction plays in intensification of a TC.

## 1 Introduction

Extreme storms like hurricanes arise from the complex interactions among the various components within the earth system. Strong winds in severe weather conditions like hurricanes result in large ocean waves, storm surges along the path of the hurricanes. In order to estimate the extent of potential risk (or impact) posed by such storms, atmosphere, ocean and surface waves are numerically modelled, where each of the system are modelled separately, by eliminating the feedback among the different systems. Within wave modelling community "wind forcing" is considered to be the largest source of error, while in atmospheric modelling, sea surface parametrization have long been considered a reason for poor forecast of storm intensity.

Studies utilising both idealised (Smith et al., 2014) and realistic model simulation (Green and Zhang, 2013) of hurricanes have demonstrated the sensitivity of the hurricane intensity to the surface layer parametrization schemes used in the model.



These parametrization schemes are used to account for exchange of momentum, heat and moisture. Although, tremendous effort has been put in improving the representation of these flux exchanges, there still exists a large degree of uncertainty in estimating these fluxes. At the ocean surface, wind waves are generated by extracting momentum from the wind, where the momentum extracted increases with the increase in wind speed. These ocean conditions where wind waves are growing is commonly referred as young sea, as opposed to the calm sea conditions or decaying sea state. Jenkins et al. (2012); Doyle et al. (2014) have shown that, in young sea conditions, ocean waves affects the effective roughness of ocean surface, which affects the wind speed, while also modulating the heat and moisture transport. Studies such as Janssen et al. (2001); Lionello et al. (2003); Warner et al. (2010) using coupled earth system model (i.e. a model where atmosphere is coupled with ocean and surface waves) have demonstrated the importance of wind-wave coupling and the role of spatial distribution of surface waves in modulating storms like hurricanes. This is because a coupled model accounts for the feedback between ocean surface waves and atmosphere, where the influence of young sea states is applied to the wind flow, subsequently affecting the wind wave generation.

Storms such as hurricanes have long been considered as heat engines (Riehl, 1963; Emanuel, 1995) fuelled by the energy extracted from the ocean, where a balance between the moist enthalpy input and the momentum dissipation is thought to be essential for the development and intensification of hurricanes. The strong winds during such severe storms causes intense breaking of ocean surface waves, resulting in generation of sea spray. This sea spray is believed to play a role in modulating air-sea flux exchanges as speculated by Ling and Kao (1976). In the last two decades, there have been numerous studies investigating the role of sea spray within the context of atmosphere-ocean interaction. These studies have focussed mainly on the effects sea spray has on momentum and heat exchange between atmosphere and ocean. Despite considerable effort, there hasn't been an unequivocal answer about the role of sea spray due to the uncertainty in modelling sea spray as well as the lack of measurements during hurricanes. The lack of measurements at high wind speeds is due to the extreme difficulty in carrying out direct measurement of air-sea thermodynamic fluxes in such conditions. The measurements of sensible heat and moisture flux are limited to wind speed of $20 ms^{-1}$. However, despite such limited observations, Andreas and DeCosmo (1999) have shown that there exist a clear signature of effects of sea spray in observed dataset. Kepert et al. (1999); Andreas and Emanuel (2001) using numerical models have shown that sea spray, when included in the model simulations can have dramatic effects on air-sea flux exchange thereby affecting storm intensity. This is because when sea spray is present, it provides an additional mechanism for exchange of heat fluxes i.e. sensible and latent heat, while also affecting the momentum exchange between the atmosphere and ocean.

When sea spray droplets are lofted in air, they increase the effective areal contact between atmosphere and ocean. In conditions prevalent during hurricanes, the ocean surface is warmer than air, therefore this enhanced areal contact between atmosphere and ocean, results in additional heat flux exchange from ocean to atmosphere. Besides the heat flux transfer from ocean to atmosphere, these sea spray droplets also extract latent heat from atmosphere so as to evaporate, thus causing some cooling in the near surface atmospheric layer. Anthes (1982) argued that during a hurricane, this strong cooling caused by the evaporating spray droplets will enhance sensible heat transfer from ocean to atmosphere, resulting in intensification of the storm. Apart from the aforementioned reasons, the evolution and impact of sea spray droplets also depends on the rate at which spray





drops are generating. Due to lack of complete knowledge of spray generation process over the wide range of droplet sizes, the flux (mass/volume) of spray droplets is usually represented with spray source generation function (SSGF). As described in (Andreas, 1998; Wu et al., 2015; Richter and Veron, 2016), there exist a number of sea spray generation functions based on the field observations, which are limited by both the number of reliable observation as well as the range of wind speed,

with no observations in hurricane intensity wind speeds. Due to the difficulty associated with field observations, studies like Fairall et al. (2009) carried out measurements in the laboratory. Even with the relative simplicity associated with laboratory environment, the results for production rate obtained from the laboratory based studies showed wide divergence thus pointing to the lack of understanding and ways to fully characterise the spray generation at the wide range of droplet sizes.

In this study, we aim to quantify the impact of coupling between a wave model and an atmosphere model during extreme

event. In order to carry out this study, we utilised a coupled atmosphere-wave model. We compare the simulation results between coupled and standalone models. In order to validate the model results, we utilise a wide range of observational datasets. Besides studying the effects of sea state on atmosphere model, we also investigate the effects of sea spray on the hurricane. For this reason, within our coupled model, an additional module for modelling the sea spray fluxes both thermal and momentum was implemented, which accounted for both atmosphere and sea state. This approach allows us to effectively

model sea spray generation, a dynamic process which is highly dependent on the sea state.

Within the context of a coupled atmosphere-ocean wave model there exist various methods for applying the effects of sea state on the atmosphere, where the most common being, recasting of sea state (from wave model) in the form of Charnock parameters (Charnock, 1955). This approach has shown improvements in the model forecasting skill. Studies like Moon et al. (2004); Hara and Belcher (2002) have proposed a more comprehensive method for coupling atmosphere-ocean wave model

using an explicit description of vertical distribution of stress within wave boundary layer. Lastly, Chen et al. (2013) have used a two dimensional description of friction velocity (specifically wave induced stress), and emphasised on the importance of the direction effects of surface waves. In the present study, we have adopted the first approach in developing a coupled atmosphere-ocean wave model, where the bulk effect of surface waves is applied.

This paper is structured as follows. In section 2, physical basis for the present approach is given, whereas section 3, provides

the description of the models, implementation of coupled atmosphere-wave model with a sea spray module and the observation data for the validation. Model setup specification, different numerical experiments are described in section 4. In the subsequent section (section 5.3), we provide the comparison of model result with in-situ measurements. Thereafter, in section 5.4, we discuss the implication of atmosphere-wave coupling, and that of sea spray fluxes on the hurricane. Finally in section 6, we summarise our results.





## 2  Background

### 2.1  Surface wave effects on atmosphere

At the air-sea interface in the atmosphere model, it is assumed that there exist two distinct layers (Janjić, 1994), first being a thin viscous sublayer over the surface and the second, a turbulent layer above it. It is assumed that the vertical transport in viscous sublayer is driven by molecular diffusion, whereas in turbulent layer, it is driven by turbulent fluxes.

As per Janjić (1994), the viscous sublayer is allowed to operate in three regimes (i) smooth and transitional (ii) rough, and (iii) rough and spray, where these regimes are distinguished based on the roughness Reynolds number $Re_r$ defined as

$$Re_r = \frac{z_0 u_*}{\nu} \tag{1}$$

Here, the roughness length $z_0$ is given by

$$z_0 = \frac{0.11\nu}{u_*} + \frac{z_{ch} u_*^2}{g} \tag{2}$$

where Charnock coefficient $z_{ch} = 0.018$, $g$ is acceleration due to gravity, $u_*$ is friction velocity and kinematic viscosity $\nu = 1.5 \times 10^{-5} m^2 s^{-1}$.

Within the wave model, following the quasilinear theory by Janssen (1989, 1991), the momentum transfer from wind to wave is defined by means of a wind input source term $S_{in}$ which account for both the sea state and wind stress. In the context of quasilinear theory, the surface roughness length $z_0$ is

$$z_0 = \frac{0.01 u_*^2}{g} \left( 1 - \frac{\tau_w}{\tau_t} \right)^{-1/2} \tag{3}$$

where $\tau_w$ is the wave induced stress and is defined as

$$\tau_w = \rho_w g \int\limits_{-\infty}^{\infty} \int\limits_{0}^{2\pi} \frac{\kappa}{\omega} S_{in} d\omega d\theta \tag{4}$$

Here, $\rho_w$ is the water density, $\omega$ is angular frequency, $\theta$ is wave propagation direction, and $\kappa$ is wave number. Furthermore, the total stress term $\tau_t$ in eq. (3) is estimated as

$$\tau_t = \rho_a u_*^2 \tag{5}$$

where friction velocity $u_* = \sqrt{C_d} U_{10}$, $\rho_a$ is air density, $C_d$ is the coefficient of drag and $U_{10}$ is the wind speed at 10 meters elevation. From eq. (3), it can be inferred that the computation of roughness length depends on the wave induced stress $\tau_w$ which in turn is calculated from the energy density spectrum, see eq. (4). For further details on the calculation procedure for $z_0$, readers are referred to MIKEbyDHI (2012); Janssen (1991).

### 2.2  Sea Spray fluxes

Following Andreas and DeCosmo (1999), it can be said that at higher wind speeds ($> 5 ms^{-1}$), within the vicinity of the ocean surface, there exists a droplet evaporation layer (DEL) which extends from the ocean surface to one significant wave height.



Within this DEL, the thermal fluxes can be separated into interfacial and sea spray mediated fluxes. Here the interfacial fluxes refer to the thermal fluxes that would exist if no sea spray influence was considered. Thus, at the top of the DEL, the total fluxes would be the combination of sea spray mediated fluxes and the thermal fluxes from the ocean surface. Also, it is further

suggested that the majority of sea spray droplets lofted in the DEL would fall back to ocean, unless they are fully absorbed or carried further aloft by the turbulent eddies, where they can act as cloud condensation nuclei. In order to investigate the effects of sea spray, it is necessary to consider their effects on both thermal and momentum flux. Here, we provide a brief description of both the thermal and momentum effects of sea spray droplets.

### 2.2.1 Thermal effects of sea spray

When sea spray droplets are lofted into air from relatively warmer ocean surface compared to air, they can exchange both heat and moisture. Also, as the sea spray are saline droplets, therefore when evaporated, they would either result in saline crystals or as suggested by Andreas (1995), attain temperature and radius, where they are in quasi-equilibrium state with their environment.

In order to model such a dynamic process, a microphysical model similar to the one suggested by Pruppacher and Klett

(1997) is needed. However, due to the complexity and excessive computation necessary to integrate such a model within a large scale atmosphere model, Andreas (1989, 1996, 2005); Kepert (1996) devised approximations to compute the equilibrium temperature and radius of the evaporating sea spray droplets. With the effects of sea spray evaporation included, the total sensible and latent heat fluxes can be written as

$$H_{L,T} = H_{L,I} + \alpha \overline{Q_L}$$
$$H_{S,T} = H_{S,I} + \beta \overline{Q_S} - (\alpha - \gamma)\overline{Q_L} \tag{6}$$

Here, $\overline{Q_S}, \overline{Q_L}$ in eq. (6) are the spray mediated "nominal" sensible and latent heat flux obtained from the microphysical calculation devised by Andreas (2005), whereas $H_{L,I}, H_{S,I}$ are the interfacial latent and sensible heat fluxes representing the interaction at air-sea interface. Also, $\alpha, \beta, \gamma$ in eq. (6) are small non-negative constants obtained by statistical fitting "nominal" fluxes to the field observations. The "nominal" fluxes in eq. (6) are obtained by integrating sensible and latent flux for all the spray droplet radius $r$ considered in the model:

$$\overline{Q_S} = \int_{r_1}^{r_2} Q_S(r)dr, \qquad \overline{Q_L} = \int_{r_1}^{r_2} Q_L(r)dr \tag{7}$$

where $r_1, r_2$ are the minimum and maximum radius of spray droplet considered in the microphysical computation. The $Q_L, Q_S$ in the eq. (7) are calculated for each droplet radius, where latent heat flux is given as

$$Q_L(r) = \begin{cases} -\rho_s L_v \left(1 - \left(\frac{r(\tau_f)}{r_0}\right)^3\right)\left(\frac{4\pi r^3}{3}\frac{dF}{dr}\right), & \text{if } \tau_f \leq \tau_r \\ -\rho_s L_v \left(1 - \left(\frac{r_{eq}}{r_0}\right)^3\right)\left(\frac{4\pi r^3}{3}\frac{dF}{dr}\right), & \text{otherwise} \end{cases} \tag{8}$$

while sensible heat flux is

$$Q_S(r) = \rho_s c_{ps}(T_s - T_{eq})\left[1 - \exp(\frac{-\tau_f}{\tau_T})\right]\left(\frac{4\pi r^3}{3}\frac{dF}{dr}\right) \tag{9}$$





In eqs. (8) and (9), $\tau_f, \tau_T, \tau_r$ represent three different time scales associated with the different stages of the sea spray droplets. As per Andreas (1990), $\tau_f$ is the time duration spray droplet remain aloft in air, while $\tau_T$ is the time taken by droplet to cool and finally, $\tau_r$ is the time taken by droplet to evaporate. $T_s, T_{eq}$ are the temperature of sea surface and the equilibrium temperature of evaporating droplets, $\rho_s$ is density of sea water, $c_{ps}$ is specific heat of sea water at constant pressure, $L_v$ is the latent heat of vaporisation, and $r_{eq}$ is the radius of droplets at when it reaches equilibrium with its environment. The term $dF/dr$ in eqs. (8) and (9) represents sea spray generation function (SSGF), which is the rate at which droplet with initial radius $r_0$ are generated

at the sea surface, while $(4\pi r^3/3)dF/dr$ is the total volume flux of spray generated at the sea surface.

Following Andreas (1989, 1990, 2005), the microphysical quantities in eqs. (8) and (9), i.e. the temperature evolution of spray droplet is approximated as

$$\frac{T(t) - T_{eq}}{T_s - T_{eq}} = \exp(-t/\tau_T) \tag{10}$$

and the radius evolution is approximated as

$$\frac{r(t) - r_{eq}}{r_0 - r_{eq}} = \exp(-t/\tau_r) \tag{11}$$

These microphysical quantities, depend not only on the initial droplet radius and air-sea temperature difference, but also on the relative humidity near the sea surface, water salinity as well as sea level pressure. For the sake of brevity, readers are referred to Andreas (1989, 1992) for details regarding the computation of these terms.

### 2.2.2  Momentum effects of sea spray

Furthermore, when sea spray is present in the DEL, total surface stress $\tau_t$ can be partitioned into ocean wave induced surface stress $\tau_w$, surface stress supported by sea spray droplets $\tau_{sp}$ and the viscous stress $\tau_\nu$ at the sea surface. The total stress $\tau_t$ in eq. (3) can be written as:

$$\tau_t = \tau_w + \tau_\nu + \tau_{sp} \tag{12}$$

In order to obtain the sea spray induced stress $\tau_{sp}$, we follow the approach similar to the one used to obtain spray induced

thermal fluxes, where we compute the contribution of individual droplets and then integrate them over the droplet radius considered so as to obtain the total spray induced stress. Following Andreas and Emanuel (2001), the spray induced stress can be written as:

$$\tau_{sp} = \frac{4\pi}{3} \rho_s \int_{r1}^{r2} u_{sp}(r) r_0^3 \frac{dF}{dr} dr \tag{13}$$

where $u_{sp}$ is the horizontal velocity of spray droplet before falling back in ocean.

### 25    2.2.3  Sea spray generation function

The sea spray droplets present in the near surface layer can be classified in two broad categories, film and jet droplets that generate by means of bursting of bubbles formed due to the air trapped by the breaking of waves, and spume droplets that





generate by means of tearing off of the wave crest. As mentioned in Andreas (1998), spume droplets are generated at higher wind speed ($> 9ms^{-1}$) and are usually of radius greater than $30\mu m$ and can be as large as $500\mu m$, whereas the film and jet droplets have radius less than $30\mu m$. Furthermore, when the spray droplets are generated, they are either at the ocean wave propagation speed or at rest. These droplets when lofted in air, gets accelerated by the wind speed, thereby affecting the

momentum exchange at the air-sea interface. Few possible explanation on the effects of sea spray on the momentum exchange have been provided in literature, where Andreas (2004) argued that the spray droplets when return to ocean surface, will result in sheltering of short waves, responsible for carrying much of the wave stress, while Bye and Jenkins (2006); Kepert et al. (1999) suggested that the presence of spray droplets will cause suppression of turbulence, due to spray droplet mass loading, and will increase the stability of the boundary layer.

From eqs. (8), (9) and (13), it is evident that the volume flux (or SSGF) of spray generated has a direct influence on the spray mediated thermal and momentum fluxes. Mueller and Veron (2014); Andreas (2004) showed that the spray effects on momentum and thermal fluxes increase at higher wind speeds, where it is implied that this is because at higher wind speeds there are large number of spume droplets present. However, most of SSGF in the literature are only valid at wind speeds below $20ms^{-1}$ and droplet radius below $30\mu m$. Following Fairall et al. (1990); Andreas (1992); Fairall et al. (1994), the spectral

distribution of sea spray droplets $S_n(r)$ can be written as

$$S_n(r) = W_f(U)f_n(r) \tag{14}$$

where, $U$ is wind speed, usually taken at 10m height, $W_f$ is the fraction of surface covered by whitecaps, and $f_n$ is the distribution of droplet spectrum. Whitecaps generated at the sea surface can be taken as representation of wave energy dissipation. Recent studies Anguelova and Hwang (2016); Scanlon et al. (2016) have described two different approaches of obtaining the

whitecap fraction, where the former follows the approach described by Phillips (1985), while latter follows the method suggested in Kraan et al. (1996), hereafter referred as "Kraan96". Following Janssen (2012); Breivik et al. (2015), the turbulent kinetic energy (TKE) flux $\phi_{oc}$ from breaking waves to ocean is related to the dissipation source function $S_{ds}$ of the spectral wave model and can be given as:

$$\phi_{oc} = \rho_w g \int\limits_{0}^{2\pi} \int\limits_{0}^{\infty} S_{ds} d\omega d\theta \tag{15}$$

where $\theta$ is wave direction. As per Kraan et al. (1996), it can be further assumed that the TKE flux $\phi_{oc}$ in eq. (15) is linearly proportional to whitecap fraction $W_f$ as

$$\phi_{oc} = \epsilon \rho_w g W_f \omega_p E \tag{16}$$

Here, $\omega_p$ is angular frequency corresponding to the peak wave energy density, $E$ is the total wave energy density, $\epsilon$ is the average fraction of wave energy dissipated per whitecap event, and is set to 0.01. In applying eqs. (15) and (16), it is implicitly assumed that in the equilibrium range (Phillips, 1985) the energy source $S_{in}$ and sink terms $S_{ds}$ in the wave action density equation (Komen et al., 1984) are in balance (Hanson and Phillips, 1999):

$$S_{in} + S_{nl} - S_{ds} = 0 \tag{17}$$





Here, $S_{nl}$ is the non-linear wave-wave interaction which represents the redistribution of wave energy from large scales to smaller scales. Hence, in the integrated action balance equation, wind input $S_{in}$ and energy dissipation $S_{ds}$ are in balance, thus permitting the usage of dissipation source function for estimating the spray mediated stress term given in eq. (12).

## 3 Methodology

The coupled modelling system used in this study consists of three components: a non-hydrostatic meteorological model (WRF), a third generation wave model (DHI MIKE21 SW) and a model coupling interface (MCI). The model coupling interface is responsible for the regridding and exchange of data between the atmospheric and ocean wave model. These components and a brief overview of the coupling methodology are described below.

### 3.1 Atmospheric Model

The atmospheric model within the coupled model is Advanced Research (ARW) Weather Research and Forecasting (WRF) version 3.4.1 (Skamarock et al., 2008). It is a non-hydrostatic atmospheric model which has been extensively used in the operational forecasts, as well as for research purpose in both realistic and ideal configurations. WRF model provides a suite of physics schemes and a variety of physical parametrizations for simulating wide range of meteorological processes.

In the present study, the outer domain in WRF model spans from the 100°W to 55°W in longitudinal and from 13°N to 45°N in latitudinal direction. The outer domain has a horizontal resolution of 21.6km and uses 41 vertical sigma levels. It covers the span shown in Figure 1. There is also a stationary nest within the outer domain, which has a horizontal resolution of 7.2km and uses 41 vertical levels. The initial and lateral boundary conditions for the atmosphere simulations were taken from Modern-Era Retrospective analysis for Research and Applications, Version 2 (MERRA-2) (Bosilovich et al., 2015) dataset, which has a resolution of 0.5°x 0.625°(50km in latitudinal direction). Due to the computational constraints, as well as the need to perform multiple simulation, we chose to apply only one-way nesting approach, where the feedback from nested domain to outer domain was turned off. The lateral boundary conditions from MERRA-2 to the outer domain were supplied at every 6h interval, while that from outer domain to the nested domain at every 1h interval. Also grid nudging was applied in the outer domain, so  for the present study, we used Simplified Arakawa-Schubert scheme (Han and Pan, 2011) for convection, Ferrier scheme (Rogers et al., 2001) for microphysics, Rapid Radiative Transfer Model (RRTM) scheme (Mlawer et al., 1997) for longwave radiation, Dudhia scheme (Dudhia, 1989) for shortwave radiation, and NOAH land surface model. The planetary boundary layer was modelled using Yonsei University scheme (YSU) (Hong et al., 2006), together with the Monin-Obukhov theory based surface layer scheme. We conducted a number of standalone simulations (not shown here) to choose the set of physics scheme which provide the best hurricane track in comparison to the best track.

### 3.2 Wave Model

The ocean wave model used is the MIKE 21 SW (Sørensen et al., 2004; MIKEbyDHI, 2012). It is a third generation spectral wind-wave model based on unstructured grid and solves the wave action density equation, where it accounts for the wave



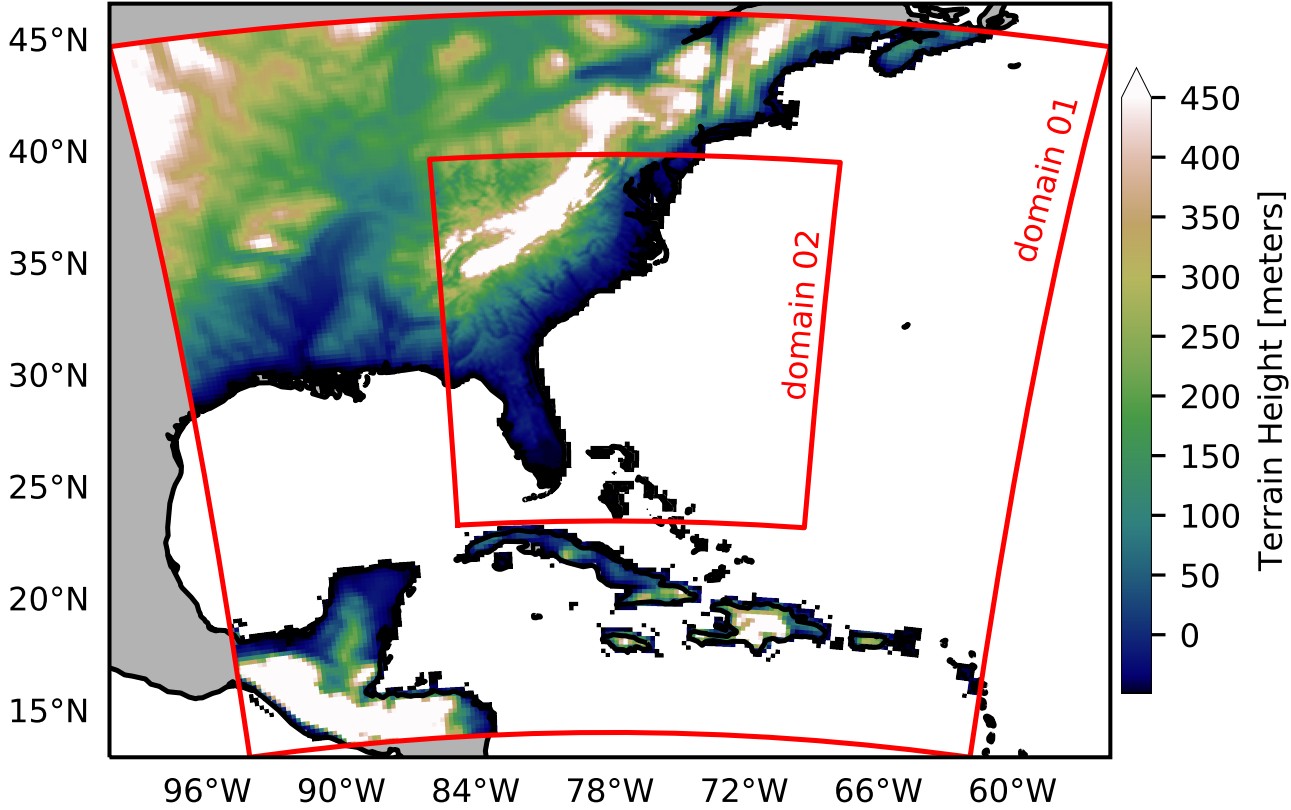

**Figure 1.** Horizontal extent and terrain elevation of atmosphere model domains in WRF, where horizontal resolution of outer domain is 21.6km while that of inner domain is 7.2km

growth, wave energy dissipation due to white-capping, bottom friction and depth as well as the non-linear wave-wave inter-
action. The spectral wave model can also account for wave-current interaction, ocean surface elevation, however these effects
were not utilised in the present study.

The grid used in the ocean wave model comprise of unstructured triangular meshes, where the outer bounds of wave domain
are within the nested domain used in the atmospheric model grid. The model setup uses 31 logarithmically spaced frequencies
(0.04-0.7 Hz) and 24 equally spaced (15°) directions.

The bathymetry (see fig. 2) used in the ocean wave model was constructed from the General Bathymetric Chart of the
Oceans (GEBCO[1]) 30 arc-second interval grid, where the bathymetric data was interpolated on to the mesh nodes. Due to
low resolution of GEBCO data, in the coastal areas data from 3 arc-second U.S. Coastal Relief Model (CRM[2]) was also

---

[1]http://www.gebco.net
[2]https://www.ngdc.noaa.gov/mgg/coastal/crm.html







**Figure 2.** Horizontal extent and bathymetry of wave model domain in MIKE 21 SW together with the extent of inner domain used in atmospheric model (red)

used. The lateral boundary condition for the wave model were obtained from the well-validated IOWAGA (Integrated Ocean Waves for Geophysical and Other Applications) (Stopa et al., 2016) global wave hindcast, conducted using WAVEWATCH-III wave model (Tolman et al., 2002). The wave hindcast was constructed using the winds from CFSR (Climate Forecast System Reanalysis) dataset (Saha et al., 2014).





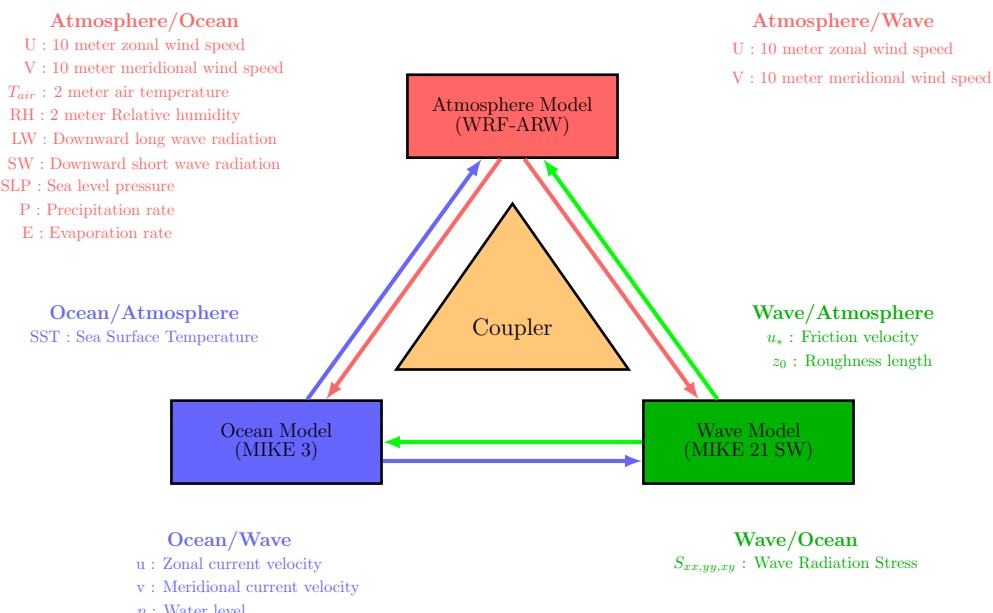

**Figure 3.** Schematics of coupled model components, where the atmosphere model (WRF), ocean model (MIKE 3) and wave model (MIKE 21 SW) interact through model coupling interface.

## 3.3   Model Coupling Interface

The model coupling interface handles the interaction between the different model components. It is used for remapping and interpolation of variable between different model components, and contains coupling physics module. The coupling physics includes sea spray model, wave boundary layer model. The schematic for different model components with respective variables used in model coupling are provided in the Figure 3. In the present study, we only utilised the atmosphere-wave coupling aspect of the coupling interface, as the present study is intended to study the effects of sea state dependent momentum and spray fluxes on TC.

Unstructured grid allows better representation of coastline features with minimal computational overhead compared to structured grids at comparable resolution, it creates disparity between the land/sea mask used by atmosphere and ocean/wave model, if the model grids (i.e. atmosphere and ocean grid) use different kind of meshes (e.g. unstructured and structured mesh). Also, the land/sea mask in the atmosphere model depends on both the model resolution as well as the land/sea mask used in the global model (from which the initial conditions have been obtained), while the land/sea mask in the ocean model is controlled by the quality of coastline. We primarily use distance weighted remapping scheme (Jones, 1998) for the data exchange. However, as a consequence of the differences in land/sea mask, at some grid locations, we also use nearest neighbour remapping (Watson, 1999) when exchanging data from atmosphere to ocean/ wave model.



### 3.3.1 Atmosphere-wave coupling

When waves are present, they affect the roughness length on the water surface, which affects the wind velocities and heat flux within the surface layer of the atmosphere. In this study, the atmosphere model provided the wind velocities at the height of 10 meter to wave model, where the wave model in turn provides the surface roughness length to the atmosphere model. It is important to point out that the MIKE 21 SW model doesn't account for the stratification of surface layer i.e. it assumes that the surface layer is neutrally stratified. In order to realise the coupling between the atmosphere and wave model, within the model coupling interface, we implemented COARE 2.6 (Fairall et al., 1996) bulk flux algorithm, which adjusts the wind velocities obtained from atmosphere model for neutral stratification.

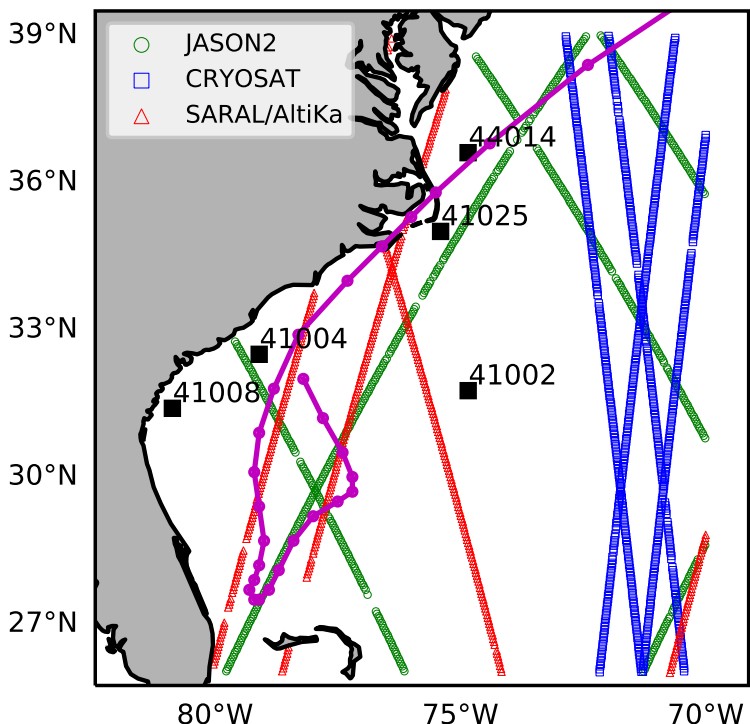

**Figure 4.** Altimeter tracks for JASON2 (green), CRYOSAT (blue) and SARAL/Altika (red) for analysis period within study region, location of considered NDBC buoys are marked in black and observed track of Hurricane Arthur (2014) is overlaid in magenta





### 3.4 Observation Dataset Sources

#### 3.4.1 Wave buoys

In this study, surface measurements of wind and wave from two NDBC[3] from the number of wave buoys distributed along the United State east coast were used (see fig. 4). The wave buoys provide measurements for wind speed, wind direction, air temperature, significant wave height ($H_s$), wave period ($T_p$), where the relative error in $H_s$ is generally predicted to be ($< 5\%$) few percent.

#### 3.4.2 Satellite data

Data for wind speed and $H_s$ from the blended product of 3 different satellites: JASON-2, CRYOSAT and SARAL (see fig. 4) were obtained from French Research Institute for Exploitation of the Sea (IFREMER). These satellites follow an orbit with a period of 10 days and provide along track data with an approximate resolution of 6 km. The dataset from the satellite altimeter measurement has some limitations as given in Cavaleri and Sclavo (2006), where the wind speed data is only reliable for 2 $\mathrm{m}s^{-1}$ to 20 $\mathrm{m}s^{-1}$, additionally the $H_s$ measurements are not reliable beyond 20 m.

## 4 Model Application

### 4.1 Synopsis of Hurricane Arthur (2014)

Hurricane Arthur was the first named storm of 2014 hurricane season. It was first identified as a tropical depression on 0000 UTC[4] July 1, 2014 by National Hurricane Centre (NHC) while it was located 70 nautical miles north of Freeport, Bahamas (Berg, 2015). It subsequently upgraded to tropical storm on 1200 UTC July 1, 2014. By then the depression has drifted westward to 60 nautical miles east of Ft. Pierce, Florida. Arthur while located in a weak mid-level steering meandered east of Florida till July 2. On July 2, a mid-level anticyclone developing over the western Atlantic caused Arthur to track northward, whereby encountering low upper level winds and warmer ocean temperature ($> 28°$) Arthur strengthened while located east of Florida. It subsequently upgraded to a hurricane on 0000 UTC July 3, 2014 located 125 nautical miles east-southeast of Savannah, Georgia.

Later that day, Arthur turned north-northeastward, accelerating while moving between a ridge over western Atlantic and a mid to upper level trough over the east United States. It continued to strengthen and reached its peak intensity of 85 knots at 0000 UTC July 4, 2014 just off the coast of North Carolina. At 0315 UTC on July 4, 2014 it made landfall just west of Cape Lookout, North Carolina. After landfall, and crossing Outer Banks, Arthur accelerated northeastward over the western Atlantic on July 4 and early July 5. It subsequently transitioned to an extratropical cyclone at 1200 UTC July 5 just west of Nova Scotia.

---

[3]http://www.ndbc.noaa.gov/

[4]Coordinated Universal Time





**Table 1.** Summary of numerical model experiments with different coupling configurations

| Experiment No. | Wave Coupling | Spray Heat Fluxes | Spray Stress |
|---|---|---|---|
| 1 | No | No | No |
| 2 | Yes | No | No |
| 3 | Yes | Yes | No |
| 4 | Yes | Yes | Yes |

### 4.2 Numerical Experiments

To evaluate the effects of ocean surface waves and ocean waves dependent sea spray on the tropical cyclone, four numerical experiments (Table 1) were conducted. In the Expt. 1, we conducted an uncoupled atmosphere model run, which is also the control run for the present study. In the uncoupled run, the surface stress was estimated using eq. (2). In Expt. 2, the effect of ocean surface waves was applied on the atmosphere model, where the surface stress was obtained from the wave model. In Expt. 3 and 4, the sea spray fluxes were added to the coupled model runs, where in Expt. 3 only the spray mediated heat fluxes were applied, while in Expt. 4, both the spray mediated heat and momentum fluxes were applied. All the numerical experiments were initialised at 0000 UTC on 30 June 2014 and integrated for 120 hours. This allows a 24 hour spin up period in atmospheric model before Arthur strengthen to tropical depression at 0000 UTC on 1 July 2014.

The model runs summarised in Table 1 do not include data assimilation, as the motivation was to investigate the response of modelling system with the inclusion of different physical processes. To assess the validity of the model results, typical error metrics of normalised bias (NBIAS), root mean square error (RMSE), Pearson correlation coefficient (R) and scatter index (SI) were used, where the model estimates are expressed as $y$ to the observed data $x$:

$$NBIAS = \frac{(\overline{y} - \overline{x})}{\sqrt{\frac{1}{n}\sum_{i=1}^{n} x_i^2}}$$

$$RMSE = \sqrt{\frac{1}{n}\sum_{i=1}^{n}(y_i - x_i)^2}$$

$$R = \frac{\sum_{i=1}^{n}(y_i - \overline{y})(x_i - \overline{x})}{\left[\sqrt{\frac{1}{n}\sum_{i=1}^{n}(y_i - \overline{y})^2}\sqrt{\frac{1}{n}\sum_{i=1}^{n}(x_i - \overline{x})^2}\right]}$$

$$SI = \frac{1}{\overline{x}}\sqrt{\frac{1}{n}\sum_{i=1}^{n}\left[(y_i - \overline{y}) - (x_i - \overline{x})\right]^2}$$

(18)

here, over bar denotes mean and $n$ is the number of observations.



**Figure 5.** Wind speed dependence of whitecap fraction (on log scale) obtained from (a) Kraan96 (Kraan et al., 1996), AH2016 (Anguelova and Hwang, 2016) and MOM80 (Monahan and Muircheartaigh, 1980), (b) As in (a), but data are binned in wind speed bins of 1 ms$^{-1}$ using data from buoy 46001

# 5  Results and Discussion

## 5.1  Spray flux parametrization

5  Following the arguments of our approach in section 2.2, we model the sea spray fluxes using the wave energy dissipation. Here, we compare the whitecap fraction obtained from wave energy spectrum with the model used in present study to that of the



model derived in recent study of Anguelova and Hwang (2016). We also obtain the constant terms needed to obtain "effective" sea spray fluxes from the "nominal" spray fluxes calculated using the microphysical model described in the section 2.2.

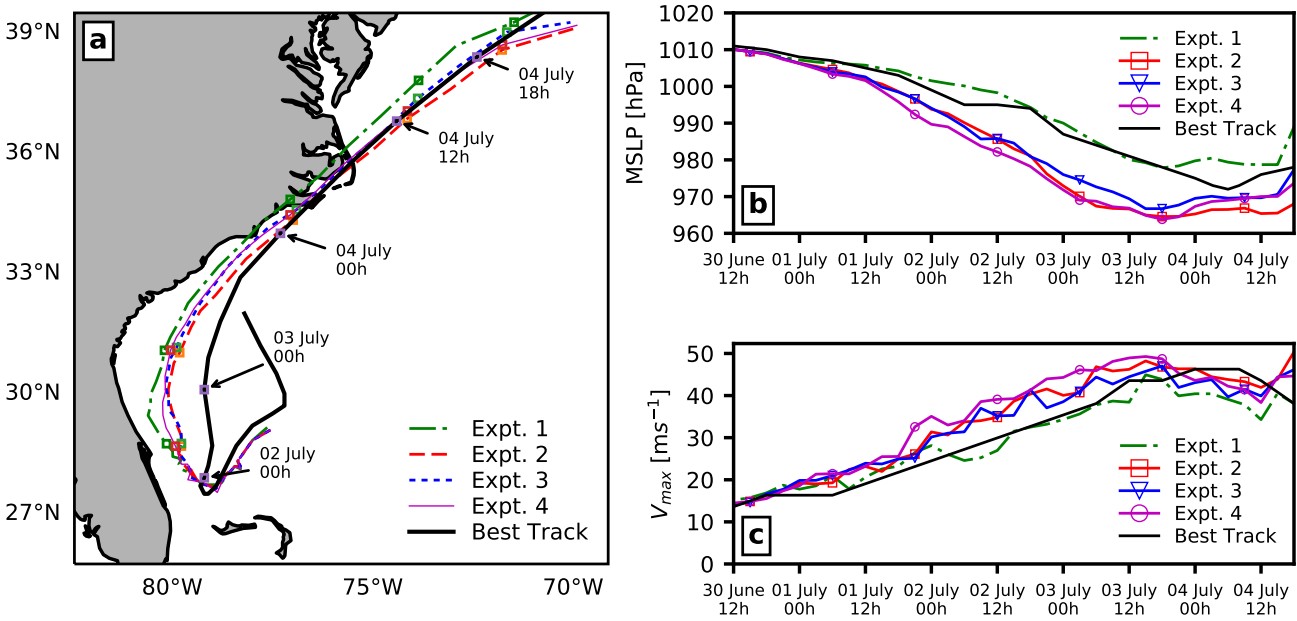

**Figure 6.** Comparison of (a) Hurricane track (b) Time series of minimum sea level pressure (MSLP), and (c) Time series of maximum wind speed ($V_{max}$) for observed and four model experiments

### 5.1.1 Whitecap fraction

As described earlier (section 2.2.3), in the present study we estimate SSGF using eq. (14), where the whitecap fraction is
5    obtained via eq. (16). Anguelova and Hwang (2016) developed a parametric model for estimating whitecap fraction, where they applied their model to wave energy spectrum observations from NDBC buoy 46001 moored in Gulf of Alaska at (56.3°N, 147.9°W). They also compared the results from their model to the photographic measurements of whitecap fraction obtained in Gulf Alaska. It was shown that the whitecap fraction obtained from their model were comparable to the photographic observation. They also compared their results with the whitecap fraction model from Monahan and Muircheartaigh (1980),
10   hereafter referred as MOM80.

Figure 5 shows the comparison of whitecap fraction obtained using Monahan and Muircheartaigh (1980); Anguelova and Hwang (2016) and eq. (16), where the results obtained from eqs. (15) and (16) are comparable to Anguelova and Hwang (2016). For details on Anguelova and Hwang (2016) model hereafter referred as AH2016, processing of wave energy spectrum from buoy measurements and their validation procedure, readers are referred to Anguelova and Hwang (2016). Figure 5(b) shows





the same data (fig. 5(a)) binned by wind speed. From Figure 5, we can infer that the whitecap fraction obtained from eqs. (15) and (16) are higher than that from AH2016 model. We can also see that both methods show similar wind speed dependence and at higher wind speed, both give lower whitecap fraction compared to MOM80 model.

### 5.1.2 Estimation of sea spray fluxes

One implication of using wave state dependent SSGF is the need to obtain the coefficient $\alpha, \beta, \gamma$ given in eqs. (5) and (6). As described in section 2.2.1, these coefficients are obtained from statistical fit of total fluxes (spray mediated and interfacial) to the fluxes from field observations. From eqs. (7) and (8) we know that the sea spray fluxes depend on the volume flux of sea spray ejected in the DEL. For the purpose of this study, we followed the procedure described in Andreas and DeCosmo (1999) and utilised the HEXOS dataset (DeCosmo, 1991). Andreas et al. (2008) utilised the same HEXOS dataset together with FASTEX dataset (Persson et al., 2005) to obtain the constant terms for the spray flux algorithm. Using microphysical spray model with the COARE 2.6 bulk flux algorithm we obtained $\alpha = 7.7036$, $\beta = 0.0$ and $\gamma = 8.3202$. We also calculated the correlation coefficients of modelled fluxes to the observed fluxes, where the correlation coefficient for sensible heat was 0.93 and for latent heat was 0.89. For the sake of brevity, we do not show the plots comparing the modelled fluxes to fluxes obtained from observation dataset. Using eqs. (5) and (6), total enthalpy flux above the DEL can be written as:

$$H_{S,T} + H_{L,T} = H_{L,I} + H_{S,I} + \beta \overline{Q_S} + \gamma \overline{Q_L} \tag{19}$$

When viewed in context of enthalpy flux, it can be said that only $\beta, \gamma$ have an effect on heat flux transfer from ocean to atmosphere. The values for the constants obtained in present study imply that only spray mediated latent flux have a role on the heat flux transfer. This is in contrast to the results obtained in Andreas et al. (2008) and Andreas and DeCosmo (1999) where they found $\beta$ to be positive non-zero value. Also, it contradicts with the conclusion that spray sensible heat flux is the primary route by which spray affects the storm energy as stated in Andreas and Emanuel (2001). However, we want to stress that further investigation with more observation data is needed to support our earlier statement.

## 5.2 Storm track and intensity

### 5.2.1 Storm track

A comparison of simulated hurricane tracks, obtained using the minimum sea level pressure, with the location of storm centre from best track and model results is presented in Figure 6(a). The simulated storm tracks are generally consistent with the best track, where the modelled storms first track southwestward and thereafter turning and move northward before making landfall. We can notice that all the modelled storms are westward of the best track, however when coupled with wave model the storm tracks improved. We can also see that the storm in uncoupled model (Expt. 1) has a higher translation speed compared to coupled atmosphere-wave model (Expt. 2-4) and best track data.

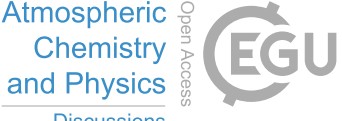

Although, sea spray coupling (Expt 3-4) do not have an appreciable effects on the model track, it does affect the translation speed of storm, where in Expt. 3 (i.e. sea spray coupling with only heat fluxes) storm moves faster compared to Expt. 4 (i.e. sea spray coupling with heat and momentum fluxes). Also both the storms moved faster compared to Expt. 2.

**Figure 7.** Comparison of coupled model results with NDBC buoys (a) 41002 and (b) 44014 for mean wind speed $U_{10}$ [ms$^{-1}$] at 10m elevation



### 5.2.2 Minimum sea level pressure (MSLP)

The time series of minimum sea level pressure (MSLP) are compared with the MSLP from best track data in fig. 6(b). Comparing the results of different numerical experiments (Expt. 1-4) indicate that uncoupled model underestimates the storm intensity while the coupled model overestimates the storm intensity. The effect of both sea spray heat and momentum flux (Expt. 4) have little effect compared to atmosphere-wave coupled model (Expt. 2). If only sea spray heat fluxes are considered (Expt. 3), the storm intensity is closer to the best track data. Including the wave and sea spray coupling (Expt. 2-4), the storm intensifies earlier compared to uncoupled model (Expt. 1).

### 5.2.3 Maximum wind speed ($V_{max}$)

The temporal development of maximum wind speed at 10 meters for the 4 experiments (Exit. 1-4) is shown in fig. 6(c). The effects of different model couplings on maximum velocity $V_{max}$ are similar to that on MSLP. We can see that hurricane under intensifies by upto $\approx 10 ms^{-1}$ in uncoupled (Expt. 1) model compared to best track data. Also, we can note that the storm intensity improves when the atmosphere is coupled with waves (Expt. 2-4). However, in contrast to section 5.2.2, from fig. 6 it is evident that the $V_{max}$ are better modelled in Expt. 2 and Expt. 4.

## 5.3 Model Validation

### 5.3.1 Wind observations

To further investigate the effects of ocean waves and sea spray, we compare the computed wind speeds with the winds measured at NDBC buoys. Figure 4 shows the location of the buoys considered in reference to the track of Arthur. We can see that buoy 41002 is located offshore while buoy 44014 along the track of Arthur. Both coupled and uncoupled model in the present study over predict the intensity of the storm at 44014 while performing well at buoy 41002 (Figure 7(a)), however we do note that the wave coupling on atmosphere results in improved timing of the storm at buoy 44014 (Figure 7(b)). We also see that the coupling sea spray (Expt. 3-4) results in lower $V_{max}$ at the buoy location, though when spray mediated momentum flux is applied together with spray mediated heat fluxes (Expt. 4), it improves both the timing as well as the buoy location. Additionally, Figure 8 compares the wind direction at buoy 41002 and 44014 obtained from the 4 numerical experiments. Furthermore, the computed statistics for both wind speed and wave parameters (i.e. wave height $H_s$ and wave period $T_{02}$) are given in Table 2. The coupling of ocean surface wave reduces (increase) the error in wind speed at buoy 44014 (41002), however when sea spray fluxes are applied, this reduction in error diminishes. Moreover, when comparing both the RMSE and correlation between wave measurements and data obtained from various numerical experiments, it can be said that the coupling of wave model improves the model results. It can also be construed from Table 2 that it is necessary to account for both spray mediated heat and momentum flux, as when only spray mediated heat fluxes are accounted, there is a noticeable reduction in correlation coefficient with increase in RMSE.



**Figure 8.** As in Figure 7, but for wind direction

### 5.3.2 Wave observations

During a hurricane, the waves are not only affected by the winds but also by the hurricane intensity and translation speed among other factors. Figures 9 and 10 show the comparison of significant wave height $H_s$ and mean wave period $T_{02}$ for the four different model experiments with the buoy measurements.

The significant wave height $H_s$ and wave period $T_{02}$ were rather well estimated when atmosphere model was coupled with waves (Expt. 2-4) compared to uncoupled model (Expt. 1). Although the size of storm induced wave fields in coupled model (Expt. 2) indicates that the storm size is bigger compared to uncoupled model and buoy measurements. Also, when sea spray



**Table 2.** Root mean square error (RMSE) and correlation coefficient (R) for mean wind speed $U_{10}$ at 10m elevation, significant wave height $H_s$ and wave period $T_{02}$ between model experiments and buoys 41002 and 44014

| | Buoy 41002 | | | | | |
|---|---|---|---|---|---|---|
| **Runs** | $\mathbf{U_{10}}[ms^{-1}]$ | | $\mathbf{H_s}[m]$ | | $\mathbf{T_{02}}[sec]$ | |
| | RMSE | R | RMSE | R | RMSE | R |
| Expt. 1 | 1.167 | 0.884 | 0.393 | 0.932 | 0.369 | 0.866 |
| Expt. 2 | 1.354 | 0.880 | 0.387 | 0.923 | 0.374 | 0.866 |
| Expt. 3 | 1.206 | 0.882 | 0.435 | 0.882 | 0.369 | 0.844 |
| Expt. 4 | 1.131 | 0.901 | 0.394 | 0.911 | 0.339 | 0.879 |
| | Buoy 44014 | | | | | |
| Expt. 1 | 3.875 | 0.798 | 0.992 | 0.829 | 0.678 | 0.870 |
| Expt. 2 | 2.979 | 0.917 | 0.518 | 0.965 | 0.503 | 0.948 |
| Expt. 3 | 3.615 | 0.792 | 0.724 | 0.895 | 0.628 | 0.909 |
| Expt. 4 | 3.618 | 0.814 | 0.645 | 0.936 | 0.507 | 0.944 |

effects are included in the coupled model, the modelled peak $H_s$ are similar to that of measurements, hurricane passes the buoy locations earlier than observed. We attribute the bias in timing of the storm passage to the translation speed of the storm, where higher translation speed can result in increased effective fetch, thereby giving higher wave heights.

We also compared the collocated significant wave heights from model experiments to the satellite altimeter derived wave heights (fig. 11). This was done by computing the closest model data point (both in space and time) to each satellite observation point. This allows us to evaluate the spatial and temporal variation of wave field due to different processes investigated here. The computed statistics of the modelled wave heights to the altimeter data are given in Table 3. When the wave effects were included in the wave model, a higher correlation and lower scatter compared to altimeter data was observed. When sea spray was included in the coupled atmosphere-wave model runs, we can see that including the spray mediated momentum flux improves the model results.

It is noteworthy there are only minor differences between the output of uncoupled and coupled model experiments at lower wave heights, however only coupled models were able to capture the higher wave heights. It is also worth mentioning that apart from modelling uncertainties, the bias in model results can arise from the differences in temporal and spatial resolution of wave model and satellite altimeter. Furthermore, for the comparison presented in Figure 11, no spatial (or temporal) smoothing of altimeter data was carried out. This is due to the use of unstructured grid in the present study for the wave model setup.





**Figure 9.** As in Figure 7, but for significant wave height $H_s$ [m]

## 5.4  Surface fluxes

Figure 12 shows the distribution of wave model dependent whitecap fraction computed from Expt. 4 at 0000 UTC July 3, 2014, where Figure 12(a) presents the spatial distribution of whitecap fraction with wind speed contours while fig. 12(b) shows a comparison of whitecap fraction obtained from wave energy spectrum to the widely used MOM80 formula. It should be noted that the results given in Figures 12 to 15 are presented on a storm relative grid with 2km spacing in radial and 1°spacing in azimuthal direction. Here, the storm relative grid were created using storm centres which corresponds to the location of

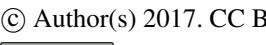



**Figure 10.** As in Figure 7, but for wave period T$_{02}$ [sec]

minimum sea level pressure. The whitecap fraction (fig. 12) and surface heat fluxes (figs. 13 to 15) obtained from wave and atmosphere model were interpolated onto the aforementioned storm relative grid. Furthermore, only data points that are within 200km from the storm centre and over the ocean are presented. From figs. 5 and 12(b), for wind speeds between $10 - 20ms^{-1}$,

5    whitecap fraction obtained from MOM80 and eqs. (15) to (17) show similar wind speed dependence, however, when extended to wind speeds present during hurricanes, whitecap fraction obtained from MOM80 are substantially higher with whitecap fraction of 1.0 at $40ms^{-1}$. This seems to show that at wind speed greater than $40ms^{-1}$ whole sea surface will be covered in whitecaps, whereas only 20% of sea surface is covered when whitecap fraction are computed from wave energy dissipation.





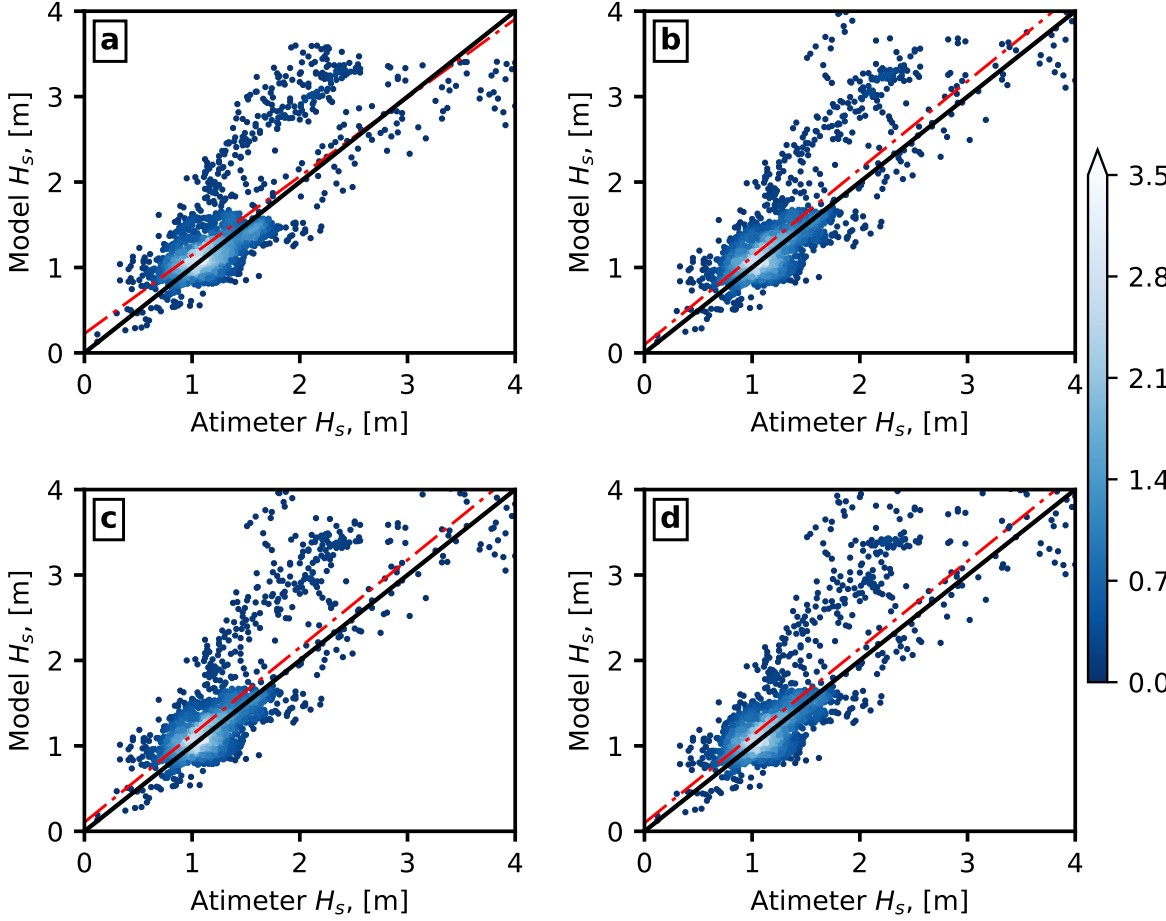

**Figure 11.** Scatter plot of modelled versus observed $H_s$ for (a) Expt. 1, (b) Expt. 2, (c) Expt. 3, and (d) Expt. 4 during June 30 to July 5 within the coupled model domain. The colorbar indicates the number of occurrence in each 0.1m bin on a logarithmic scale. The solid black represents the ideal fit of model and measurements while the dashed red line represents the linear fit of the data.

Until now, we have only discussed the wind speed dependence of whitecap fraction, when looking at the spatial distribution of whitecap fraction in relation to the wind speed (see fig. 12), it can be noted that the peak of whitecap fraction is in the rear left quadrant of the hurricane translation direction (see black arrow in fig. 13(d)), whereas the peak of wind intensity is in the front right quadrant of the hurricane. This shows that within the coupled model used in present study, not only the volume of sea spray droplets generated was altered, but also the spatial spread of sea spray volume flux was modified. This is noteworthy, as when sea spray production is parametrised using MOM80 (where whitecap fraction $W_f = 3.8e^{-6}U_{10}^{3.41}$), the peak of whitecap fraction will be collocated with the peak of wind speed. It is important to keep in mind that the results presented in fig. 12(b)





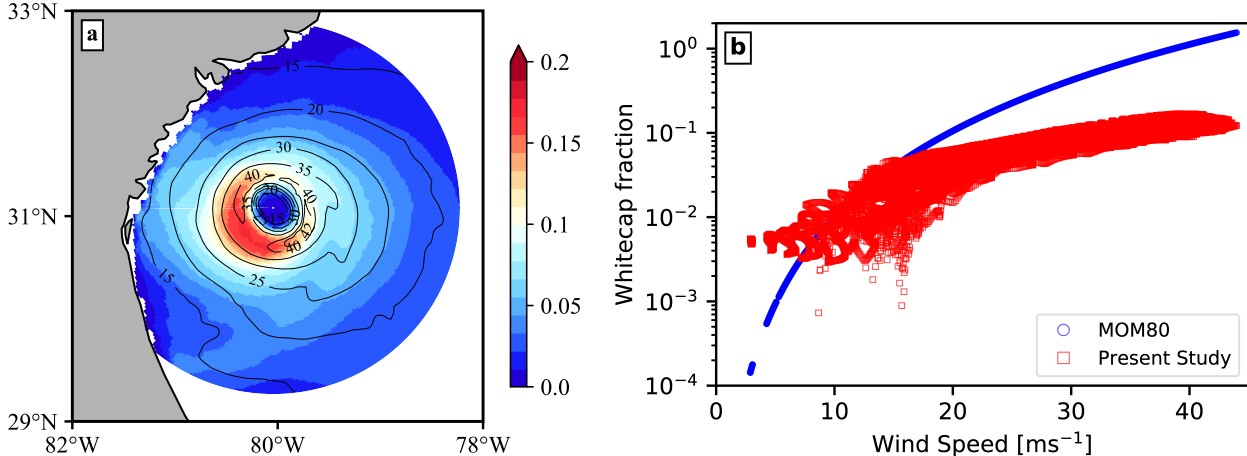

**Figure 12.** Computed whitecap fraction from Expt. 4 at 0000 UTC July 3, 2014 (a) Distribution of whitecap fraction with contours of mean wind speed $U_{10}$, (b) comparison of whitecap fraction (on log scale) versus mean wind speed $U_{10}$

**Table 3.** Statistical comparison of altimeter and wave model derived significant wave height $H_s$; normalised bias (NBIAS), root mean square error (RMSE), Pearson correlation coefficient (R) and scatter index (SI)

| Runs | NBIAS[$m$] | RMSE[$m$] | R | SI |
|---|---|---|---|---|
| Expt. 1 | 0.084 | 0.437 | 0.781 | 0.419 |
| Expt. 2 | 0.093 | 0.427 | 0.821 | 0.405 |
| Expt. 3 | 0.095 | 0.441 | 0.810 | 0.419 |
| Expt. 4 | 0.087 | 0.428 | 0.817 | 0.409 |

merely highlight the fact that in most studies MOM80 model is applied beyond the range of its validity, as is the case in fig. 12(b).

The effects of including spray mediated heat fluxes as well as ocean surface waves on the enthalpy fluxes (i.e. sensible and latent heat flux) are shown in Figures 13 to 15. Here, we first compare the total sensible heat flux (fig. 13), then the total latent heat flux (fig. 14) and eventually their azimuthal averaged radial variation in fig. 15. The heat fluxes are presented at 0000 
UTC July 3, 2014, because the storm centres were collocated (see Figure 6(a)). It is evident from the comparison of sensible and latent heat flux obtained from uncoupled atmosphere model (Expt. 1) and ocean wave coupled atmosphere model (Expt. 2), that coupling ocean waves with atmosphere results in substantial increase in sensible and latent heat fluxes. Using figs. 13 and 14, it can be argued that the increase in heat fluxes is largely due to wave induced surface roughness, rather than any air-sea temperature difference that might arise due to the usage of fixed sea surface temperature (SST) field. These results are inline



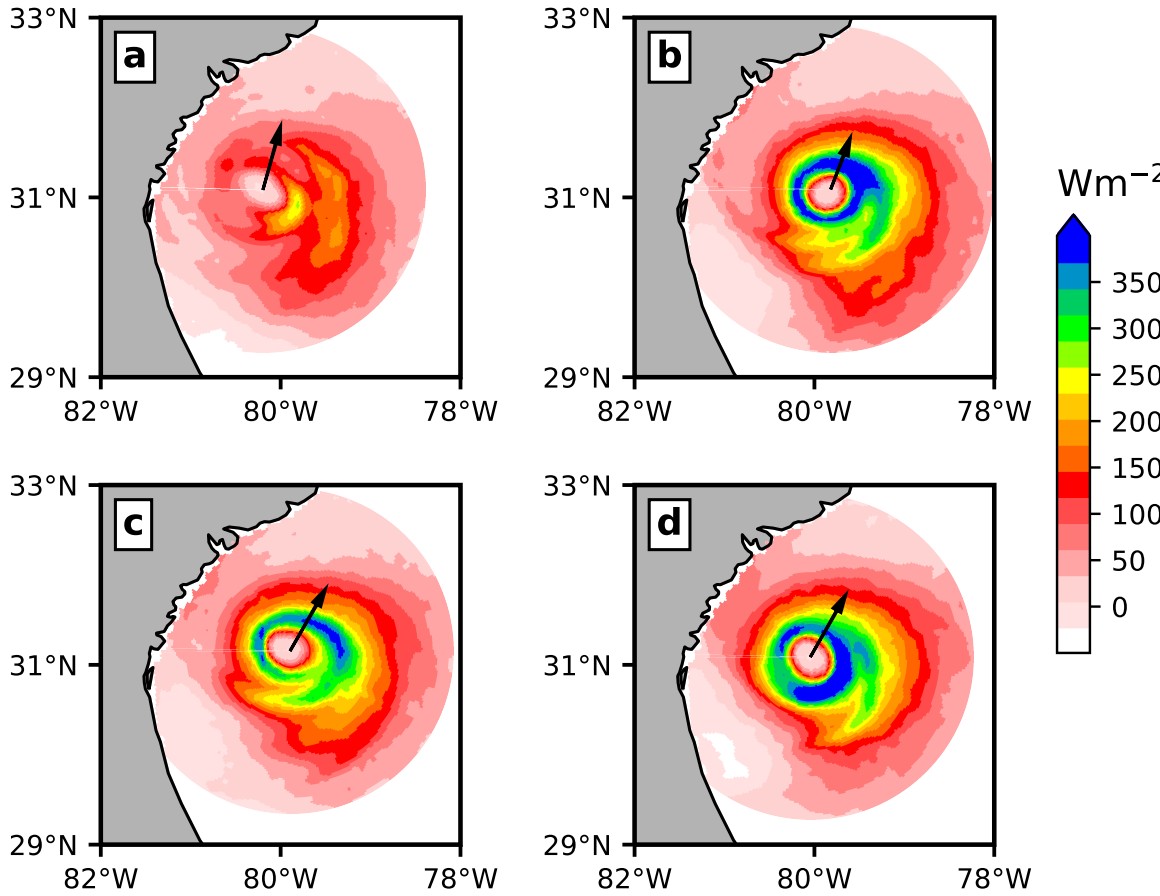

**Figure 13.** Plan views of surface sensible heat flux for (a) Expt. 1, (b) Expt. 2, (c) Expt. 3 and (d) Expt. 4 at 0000 UTC July 3, 2014. The black arrow indicates the hurricane translation direction over three hour interval.

with the arguments given in Janssen et al. (2001), where it was suggested that the increased surface roughness will enhance the surface heat fluxes causing vortex stretching, thus intensifying the storm.

By comparing fig. 13(b) and fig. 13(c), it can be noticed that when only spray mediated heat fluxes (Expt. 3) are added, there is a reduction in sensible heat flux as well as broadening of storm core compared to Expt. 2. Furthermore, when both spray mediated heat and momentum flux are added (Expt. 4), it results in higher sensible heat flux (see fig. 13(d)) compared

to Expt. 3. Besides the differences in sensible heat flux, there also exist differences in location of maximum sensible heat flux with respect to storm centre. Contrary to assumption that applying spray mediated heat flux will intensify the hurricane, these results show that the interaction between sea spray and hurricane are rather more intricate, where both the thermodynamic and dynamic processes play different roles. For instance, coupling waves with atmosphere model increases the surface roughness,





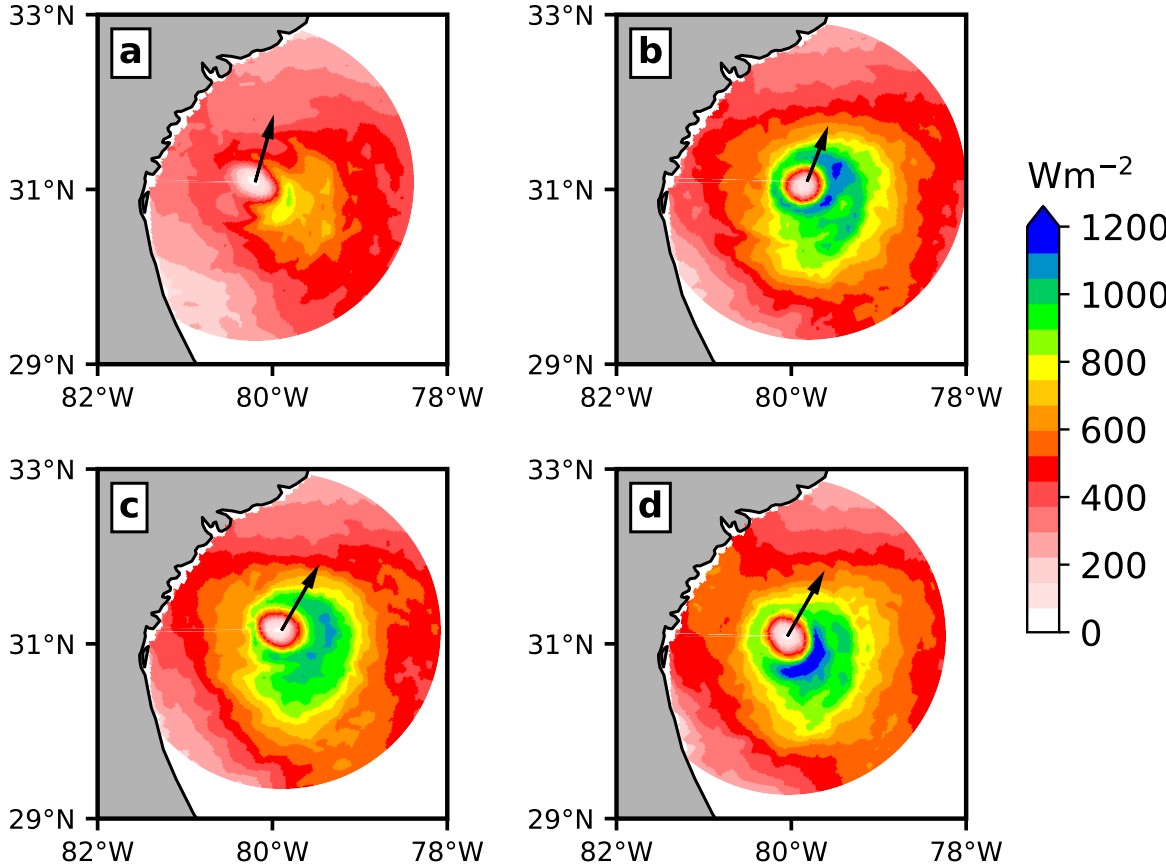

**Figure 14.** As in Figure 13, but for surface latent heat flux

which results in intensification of hurricane. Increased surface roughness may however also decelerates the hurricane as seen
5    in Figure 6, causing it to stay on warmer ocean for longer duration.

    The radial distribution of azimuthally averaged total latent and sensible heat fluxes for the 4 model experiments are presented
in Figure 15. It is clearly noticeable that in all the experiments, maximum value of heat fluxes (i.e. sensible and latent heat
flux) are in high wind region (i.e. radius = 20 to 75 km). Also, maximum value of latent heat flux in Expt. 2-4 is twice that
of Expt. 1 whereas maximum value of sensible heat flux in Expt. 2 and Expt. 4 is thrice, while in Expt. 3 is 2.5 times that of
10   Expt. 1. Besides the effects of coupling wave model with atmosphere model (Expt. 2), the effects of sea spray on sensible and
latent heat fluxes can also be noticed. In the case of sea spray heat fluxes (Expt. 3), there is a noticeable reduction in maximum
value of sensible and latent heat fluxes compared to Expt. 2. However when both spray mediated momentum and heat fluxes
are considered (Expt. 4), there is a reduction in maximum latent heat flux while an increase in the maximum sensible heat flux
compared to Expt. 2. It should also be pointed out that these difference in heat fluxes (between Expt. 2-4) are only in high wind





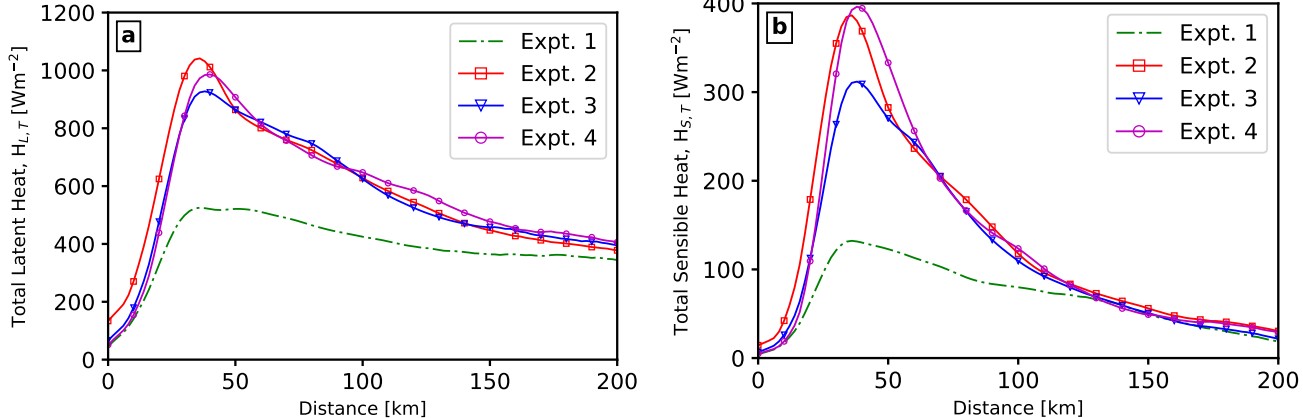

**Figure 15.** Radial distribution of azimuthal averaged total (a) latent heat flux (b) sensible heat flux at 0000 UTC July 3, 2014

region, with negligible effects at higher radius. In addition to affecting the value of heat fluxes, sea spray (Expt. 3-4) also cause slight broadening of core size compared to Expt. 2.

To examine the effects of coupling wave model and sea spray on the vertical structure of hurricane, Figure 16 shows the azimuthally averaged radius-height cross-section of temperature (fig. 16(b,d,f)) and mixing ratio $q$ (fig. 16(a,c,e)) anomaly. The radius-height cross-section utilises the storm centre located at the surface for all the vertical levels so as to construct a storm relative grid. The anomaly fields were calculated by subtracting azimuthally averaged fields for Expt. 1 from those for Expt. 2-4 respectively. For Expt. 2 (fig. 16(b)) a strong positive anomaly extends from z = 4 to 16 km and from r = 0 to 140 km, whereas a weak negative anomaly in the near surface region, at radii greater than 40 km. The warming in the upper air region within the hurricane core (i.e. near the eye wall) in Expt. 2 can be attributed to the increase in surface heat fluxes. Comparing Figure 16(b,d), we can see that, when sea spray mediated heat fluxes are added, there is a reduction in upper level warming in core region whereas enhanced cooling in near surface layers at radii greater than 4 km. Also, broadening of warm anomaly in the core can be noted, where the edge of warm anomaly in fig. 16(b) has shifted rightwards compared to the fig. 16(a). This broadening of warm anomaly and the increased cooling in near surface layers can be associated with the decreased storm intensity. Figure 16(d) shows the temperature anomaly when the spray mediated momentum flux is added together with the spray heat fluxes. The first key effect is the enhancement of the warm anomaly in the upper levels compared to Expt. 3. Although the broadening of core is still present, there has been a slight reduction in cooler region compared to the Expt. 3.

Figure 16(a) shows the mixing ratio anomaly for Expt. 2 relative to the Expt. 1 (i.e. uncoupled atmosphere model run). A broad region of positive anomaly can be noted extending from radius = 60 km to 160 km. Also, just above this region at radii greater than 120 km, a large region of negative anomaly can be noticed. The largest positive anomalies are in the near surface region in all the experiments, with the maximum occurring in the core region. When the spray heat fluxes are added (Figure 16(b)), the negative $q$ anomaly has shifted from radius = 120 km to 80 km, however with a considerable downward shift





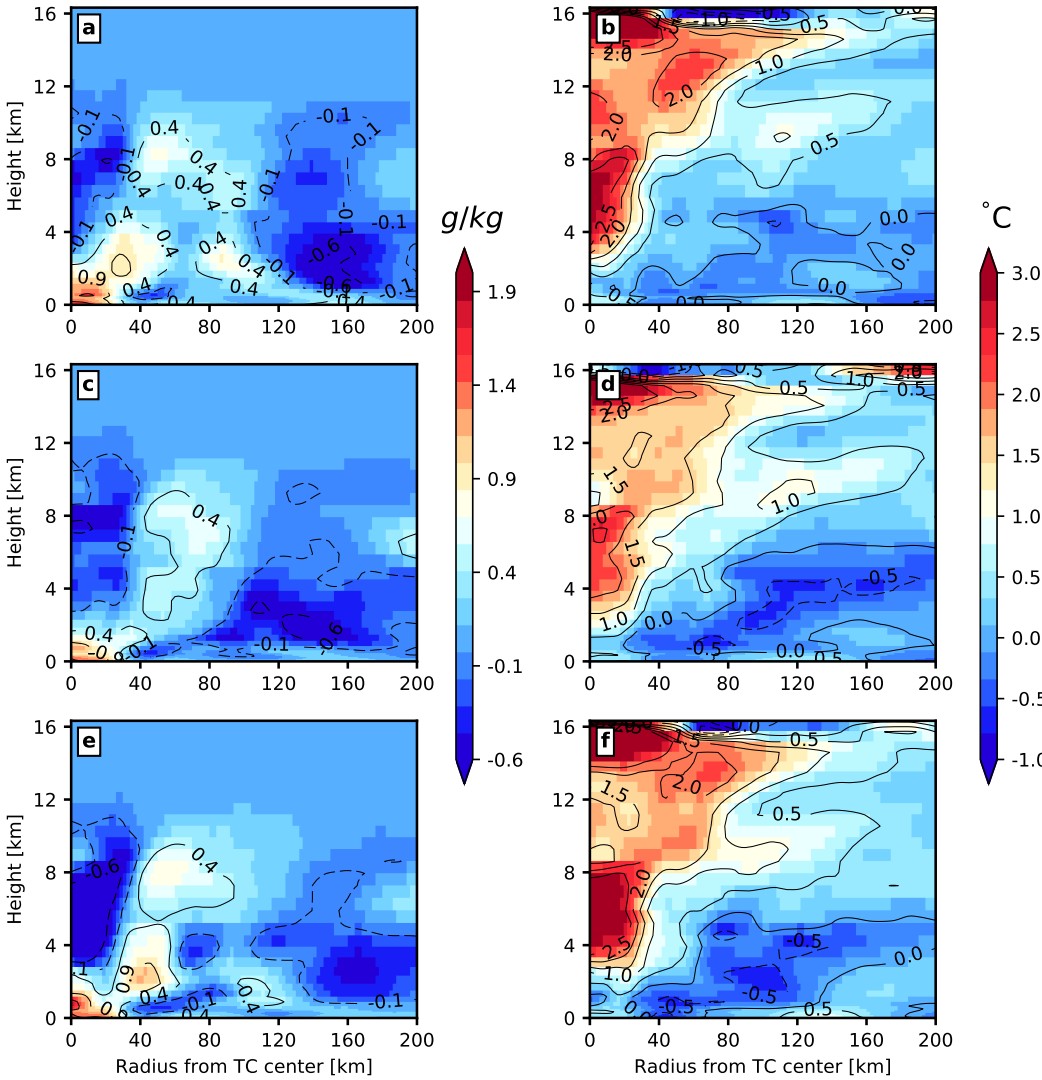

**Figure 16.** Height-radius cross sections of difference of azimuthally averaged mixing ratio (left) and temperature (right), for (a,b) Expt. 2, (c,d) Expt. 3 and (e,f) Expt. 4. The difference was calculated by subtracted azimuthal averaged quantity for Expt. 1 from respective coupled model result at 0000 UTC July 3, 2014

of the vertical extent from 12 km to 6 km. However, when both the spray heat and momentum fluxes are utilised (Figure 16(c)), the extent of negative anomaly has shifted back to 120 km radial location from 80 km. Also, worth noting is the increase in negative $q$ anomaly within the eye wall region, where the $q$ values have decreased by -0.6 g kg$^{-1}$ compared to -0.1 g kg$^{-1}$ seen in Figure 16(a,b).

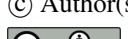



## 6   Summary

This study investigated the effects of air-sea interaction on the lifecycle of a Hurricane Arthur (2014) that traversed through

north Atlantic ocean, made landfall in north Carolina, then re-emerged over western Atlantic and eventually underwent transition to an extra-tropical storm. More specifically, this study explored the role of ocean surface waves and sea spray mediated heat and momentum fluxes on the structure and intensity of the aforementioned tropical cyclone.

There have been limited work in assessing the effects of sea spray mediated fluxes using a coupled atmosphere-wave model, where the sea spray generation was modelled using wave energy dissipation. Furthermore, most of the previous studies have

used bulk approximations of sea spray fluxes when used in conjunction with atmosphere or a coupled atmosphere-wave model. The aforesaid bulk approximations were formulated as a function of surface wind speed or friction velocity. In the present study, a consistent approach for incorporating sea spray fluxes without relying on bulk approximations was presented. Moreover, a comparison of whitecap fraction obtained from wave energy dissipation to the widely used MOM80 (Monahan and Muircheartaigh, 1980) and recently formulated AH2016 (Anguelova and Hwang, 2016) was presented. It was shown that the

method adopted in present study results in whitecap fraction comparable to the results reported by Anguelova and Hwang (2016), while resulting in substantially lower whitecap fraction at higher wind speed compared to MOM80. Due to the limitations in the sea spray microphysical model, a new set of coefficients for incorporating nominal spray fluxes using the HEXOS dataset were obtained.

To investigate the role of surface waves and sea spray fluxes, a two-way coupled atmosphere-wave model was utilised. The

coupled model was developed using a flexible coupler where, different processes (such as sea spray physics) were integrated at the coupler level. Within the coupled model, sea spray fluxes were incorporated as discrete contribution of a spectrum of spray droplets, where the spray droplet generation was modelled using the ocean wave energy dissipation due to whitecapping. The results from four different model simulations were analysed to elucidate the effects of wave induced surface roughness and spray mediated heat and momentum fluxes on the distribution of sensible and latent heat as well as the temperature and mixing

ratio among different model coupling scenarios. Furthermore, the wave model results from different numerical experiments were compared with the measurements obtained from floating offshore buoys and satellite altimeter.

As illustrated in fig. 6, the model employed in present study captures the lifecycle of simulated TC relatively well, where the uncoupled atmosphere model results in somewhat weaker TC while the coupled model results in somewhat stronger TC compared to the best track data. Furthermore, all the simulated TCs traverse westward of the best track data, however when the

atmosphere model is coupled with a surface wave model, the TC track shifts east of the uncoupled model track. Also, compared to uncoupled atmosphere model, the coupled model simulated storms are able to attain the maximum velocity similar to that of best track data but all the simulated storms attain maximum intensity almost 12 hours before that observed in best track data. Despite all the foregoing differences, it behoves us to argue that the numerical experiments performed in the present study are adequate for conducting a preliminary investigation of the role of ocean waves and sea spray fluxes.

Moreover, in the recent literature, a number of reasons have been associated with the bias in the simulated TC tracks. These include the effects of dataset used for model initialisation, initialisation time of the model run, atmosphere model resolution, and





physics scheme used for cumulus parametrization. All of these are the topics of active research, where a number of advanced techniques such as ensemble Kalman filter and advanced data assimilation have been developed to alleviate the effects of

5    initial condition error, TC bogussing for reducing the bias due to initialisation time, and Large Eddy Simulation (LES) and super-parametrization are being used to improve the cumulus parametrisation in numerical models. It is arguable that running the same numerical experiments with different initial conditions (e.g. ERA-Interim or GFS Final analyses) would be useful to assess the validity of the results presented here, however such tests are beyond the scope of present study, but recommended for future work.

     Including the sea state dependent surface roughness increases the sensible and latent heat flux exchange within the surface

layer. As the present study, does not couple an ocean model, therefore this increase in surface heat fluxes is presumably from the increased surface friction velocity. Moreover, in the present study, the sea spray fluxes (or SSGF) depend on the wave energy dissipation due to whitecapping, therefore it is not possible to distinguish between the effects of coupling sea spray fluxes and ocean surface waves. However, the results presented here do underscore the significance of the friction velocity in modulating the storm intensity. Furthermore, the results presented here also alludes to the uncertainty associated with the inclusion of sea

spray fluxes, where the limitations are due to the lack of observation data at higher wind speeds and the limited understanding of the underlying physical processes necessary for modelling sea spray fluxes.

*Acknowledgements.* Authors would like to thank DHI Water and Environment Pte Ltd for providing MIKE software package used in the present study. The authors also acknowledge the support of Energy Research Institute (ERI@N) for providing the computing resources utilised in the present study.

*Competing interests.* The authors declare that they have no conflict of interest.



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
