# Peer review of "The Effects of Sea Spray and Atmosphere-Wave Coupling on Air-Sea Exchange during Tropical Cyclone"

_Atmospheric Chemistry and Physics, 2017_

## Short Comment (SC1) · 9 Oct 2017

The paper investigated the effects of ocean surface waves and sea-spray mediated heat and momentum fluxes on the structure and intensity of TC. Due to the complexity, the ocean surface waves and heat and momentum fluxes have not yet properly modeled in the TC simulation even though tremendous efforts have be devoted into this research field. This paper proposes a 2-way coupled atmosphere-wave model to approximate the heat and momentum fluxes without relying on the conventional bulk approximations (in terms of wind speed and friction velocity, which are uncertain to some extent.). To execute the 2-way modeling, a model coupling interface is devel-

oped to link WRF with a wave model. The formulation of ocean surface waves, heat and momentum fluxes and the information exchange between different computing platforms are clearly presented. The simulation results have proven the proposed method is working and improves the accuracy of TC simulation. All in all, I find this paper timely and useful for my research study.

---

## Referee Comment (RC1) · C. Fairall (Referee) · 27 Oct 2017

This paper addresses the issue of sea spray effects using a coupled atmosphere-wave model. The advantage of using a wave model is that sea spray production can be linked directly to the wave properties. The effects of sea spray are implemented for momentum, sensible heat, and latent heat fluxes. The spray production parameterization is a fixed droplet spectrum scaled by whitecap fraction (this follows most approaches in the literature). The major advance in this paper is the use of Anguelova and Hwang's whitecap parameterization that is based on wave properties (taken from the wave model). The thermodynamic effects are implemented following the Andreas

approach. A net droplet heat flux is produced by integrating evaporation rates over the spray flux spectrum. Feedback effects also follow Andreas by fitting the difference in the COARE2.6 model vs observations from the HEXOS field program. The approach is tested via hindcasts of Hurricane Arthur.

I think the authors are basically on the right track. The results for whitecaps are much more credible than the old MOM80 formulation. * One major flaw of this work is the use of the Andreas approach for sea spray. It is based on questionable assumptions – COARE2.6 is purely interfacial so any residual is due to spray. Furthermore, the data used to arrive at the coefficients has essentially no observations past 20 m/s. Thus the actual differences in observed vs COARE2.6 fluxes is mostly noise. The formulation of the feedback coefficients might be ok, but the values are non-physical. There are more physically based formulations of feedback – Bao et al 2011 or Mueller and Veron 2014. So the results given here lack credibility. *Another thing that is not explained is the difference between Expt1 and 2. Why does this change lead to a stronger hurricane. On 26 they claim the change from 1 to 2 'increases the surface roughness, which results in intensification of the hurricane'. I find that strange – it is the opposite of conventional wisdom and needs to be explained. Also, the modest increases in windspeed lead to a doubling of sensible and latent heat flux (fig 15) for Expt 1 to 2. I don't understand that. This aspect needs a lot more explanation. *The examination of the sensitivity to spray focusses on heat fluxes, which have never been measured in hurricanes and therefore can't be verified. I suggest they could focus on the near-surface air temperature and humidity, which have been measured (e.g., we Zhang et al. 2017). *Comparisons of sensible and latent heat flux for Expts 2,3,4 suggest that spray has negligible effect on the thermodynamics. However, it is clear from fig 12b that much less spray is produced by the author's model compared to MOM80 in Andreas. I think it would be interesting to see a comparison of the total spray mass flux as a function of wind speed (this paper vs MOM80). Perhaps these could be compared to the laboratory data of Suslow et al. 2016. *On a more editorial subject, I think the description of whitecap fraction and spray function parameters (sections

5.1.1 and 5.2.1) should be moved to section 2 since they are not part of the hurricane simulations.

So, based on this I cannot recommend publication of this paper in its present form. I think it needs a lot of work.

---

## Referee Comment (RC2) · Dr Troitskaya (Referee) · 1 Nov 2017

The paper is an effort to demonstrate the effect of waves and sea-spray on development of a tropical cyclone within a coupled wind-wave model. The similar estimations for the effect of waves or spray are known from literature. This paper combines wind, waves and spray all together basing on recent models of coupling processes. The model presented in the paper can be better considered as the description of the instrument for investigation effects of coupling in air sea boundary layer, than a piece of study elucidating physical processes of the coupling. The main problem of this kind of modeling is strong uncertainty in elements of the models namely, 1) uncertainty of the

drag coefficient of the sea surface at hurricane wind speed; 2) poor quality of modeling surface waves at hurricanes (for example, DIA approximation widely used for nonlinear 4-wave coupling in wave field gives strong errors at hurricane winds); 3) uncertainty of the spray generation function given by different authors is about 3 orders of magnitude. 4) whitecap fraction is also strongly uncertain. 5) uncertainty in coefficients ïĄą, ïĄć and ïĄğ describing the feedback effect of spray-mediated fluxes.

And the authors understand well the existing uncertainties, especially in the sea-spray physics. Investigating the sensitivity of the model to the uncertainties in its elements will help to understand the importance of this issues. 1) The model demonstrates strong sensitivity to the presence of waves. First, this fact needs explanation from the physical point of view. Second, investigation of sensitivity the wave model is strongly desirable, in particular sensitivity to the wind source term and wind dissipation term. 2) Authors' estimations of the thermodynamic feedback coefficients ïĄą, ïĄć and ïĄğ are in significant contract with another papers (eg., Andreas et al, 2008; Mueller, Veron, 2014). In this connection sensitivity study to the thermal feedback coefficients is needed. 3) Sensitivity study to spray generation function is also desirable.

Explanation of the effect of spray on momentum exchange is to brief. An equation for the horizontal velocity of spray droplet before falling back in ocean in Eq.(13) is needed. Besides, the starting lines of paragraph 2.2.3 consists of qualitative discussion of the effect of spray on momentum exchange in atmospheric boundary layer, but no quantitative description is given. It definitely should be explained in some details. Possibly some related references should be added:

Bao J-W, Fairall CW, Michelson SA, Bianco L (2011) Parameterizations of sea-spray impact on the air–sea momentum and heat fluxes. Monthly Weather Rev 139:3781-3797 Kudryavtsev VN (2006) On the effect of sea drops on the atmospheric boundary layer. J Geophys Res 111:C07020. Kudryavtsev VN, Makin VK (2007) Aerodynamic roughness of the sea surface at high winds. Boundary- Layer Meteorol 125(2):289-303 Kudryavtsev V, Makin V (2011) Impact of ocean spray on the dynamics of the

ACPD
marine atmospheric boundary layer. Boundary-Layer Meteorol 40 (3):383-410 Makin VK (2005) A note on drag of the sea surface at hurricane winds. Boundary-Layer Meteorol 115 (1):169-176 Troitskaya Yu.I., Ezhova E.V., Soustova I.A., Zilitinkevich S.S. 2016: On the effect of sea spray on the aerodynamic surface drag under severe winds. Ocean Dynamics, V. 66, P. 659-669

Summing up, I recommend major revision of the paper to address the above issues.
* * *

---

## Author Comment (AC1) · 31 Jan 2018

C. Fairall (Referee)

chris.fairall@noaa.gov

1. The major advance in this paper is the use of Anguelova and Hwang's whitecap parameterization that is based on wave properties (taken from the wave model)

Ans: We do not use the whitecap parameterisation described in Anguelova and Hwang (2016), rather, we use the parameterisation for estimating whitecap fraction given in Kraan et al. (1996).

2. One major flaw of this work is the use of the Andreas approach for sea spray. It is based on questionable assumptions – COARE2.6 is purely interfacial so any residual is due to spray. Furthermore, the data used to arrive at the coefficients has essentially no observations past 20 m/s. Thus the actual differences in observed vs COARE2.6 fluxes is mostly noise.

Ans: In our study, we used COARE2.6 as Andreas's model uses COARE2.6 also. Also, this parametrization has been used in numerous publications by Andreas and his co-authors. We do understand reviewer's concern, however, we would like to mention that this is in-line with the standard practice. Besides using HEXOS data (DeCosmo, 1991), we have also conducted further tests (not presented in the paper) using FASTEX data (Persson et al., 2005) and found little effect on the value of coefficients.

Furthermore, as described in Andreas et al. (2008), we note that although the Tropical Ocean-Global Atmosphere Coupled Ocean–Atmosphere Response Experiment (COARE) version 3.0 bulk flux algorithm (Fairall et al., 2003) has been tuned with flux data collected in wind speeds up to 20 m/s and is therefore operationally useful in this wind speed range, however, as argued by Andreas et al. (2008), it is based strictly on interfacial scaling and thus may not be reliable if it is extrapolated to wind speeds above 20 m/s. Andreas (2010) argues that the version 2.6 of COARE algorithm is used because its calculations of temperature ($z_T$) and humidity ($z_Q$) roughness lengths, which are used in computing sensible heat flux ($H_S$) and latent heat flux ($H_L$), are based on surface renewal theory of Liu et al. (1979). Because this algorithm is theoretically based and proven to be accurate for treating the interfacial sensible and latent heat fluxes in winds up to 10 m/s (e.g. Fairall et al. (1996), Grant and Hignett (1998)), it should still be accurate when extrapolated to higher wind speeds.

3. The formulation of the feedback coefficients might be ok, but the values are non-physical. There are more physically based formulations of feedback – Bao et al 2011 or Mueller and Veron 2014. So the results given here lack credibility.

Ans: We do concur with the reviewer that these values need to be revisited, however, due to the lack of measurements data in wind speed beyond 18m/s, it is rather difficult at this point of time. Furthermore, we would like to add that these coefficient terms are used due to the uncertainties

associated with the sea spray generation function, and the values of these coefficient terms hold little meaning, besides the fact that they are used as correction for the "nominal" sea spray fluxes obtained from the microphysical model. The values used in Bao et al. (2011) and Mueller and Veron (2014) are also equally non-physical because of the aforementioned reason.

4. Another thing that is not explained is the difference between Expt1 and 2. Why does this change lead to a stronger hurricane. On 26 they claim the change from 1 to 2 'increases the surface roughness, which results in intensification of the hurricane'. I find that strange – it is the opposite of conventional wisdom and needs to be explained. Also, the modest increases in windspeed lead to a doubling of sensible and latent heat flux (fig 15) for Expt 1 to 2. I don't understand that. This aspect needs a lot more explanation.

Ans: We would like to bring reviewer's attention to Table 1, where the differences between the different experiments have been summarized. Furthermore, we describe the different experiments in Section 4.2. With respect to the conventional wisdom, it can be argued that there is some evidence that increasing coefficient of drag (or roughness length) results in intensification of hurricane. There are some studies investigating idealized hurricanes, where it was found that up to a certain value of coefficient of drag, the hurricane intensifies. The aforementioned results are reported by Montgomery et al. (2010), Kilroy et al. (2017). The increase in sensible and latent heat flux in our view is also due to the formulation of the heat flux parameterization used in the present study. This can be further realized from the parameterization of momentum and heat fluxes used in WRF model. The bulk formulas for momentum, sensible heat and latent heat can be written as:

$$\tau = \rho \overline{(w'u')} = \rho\, C_d (U_{ref} - U_0)^2$$

$$H_s = \rho\, c_p \overline{(w'\theta')} = \rho\, c_p C_h U_{ref} (\theta_{ref} - T_s)$$

$$H_l = \rho\, L_v \overline{(w'q')} = \rho\, L_v C_q U_{ref} (q_{ref} - q_s)$$

where $\overline{(w'q')}, \overline{(w'\theta')}, \overline{(w'u')}$ are covariance terms for humidity, potential temperature and velocity respectively. These covariance terms are more commonly written using friction velocity $u_*$ where the terms for humidity, potential temperature and velocity become $u_* q_*, u_* \theta_*, u_*^2$ and using Monin Obukhov similarity theory, the terms $q_*, \theta_*, u_*$ are given as

$$q_* = \frac{\kappa (q_{ref} - q_s)}{\ln\left(\frac{z_{ref}}{z_Q}\right) - \psi_h\left(\frac{z_{ref}}{L_0}\right)}$$

$$\theta_* = \frac{\kappa (\theta_{ref} - T_s)}{\ln\left(\frac{z_{ref}}{z_T}\right) - \psi_h\left(\frac{z_{ref}}{L_0}\right)}$$

$$u_* = \frac{\kappa U}{\ln\left(\frac{z_{ref}}{z_0}\right) - \psi_m\left(\frac{z_{ref}}{L_0}\right)}$$

here $z_0, z_T, z_Q$ are the roughness length for momentum, temperature and humidity. Now, using above equations and for neutral stratification, the coefficient of drag, heat and humidity can be written as

$$C_{d,N} = \left( \frac{\kappa}{\ln\left(\frac{z_{ref}}{z_0}\right)} \right)^2$$

$$C_{h,N} = \frac{\kappa\, C_{d,N}^{1/2}}{\ln\left(\frac{z_{ref}}{z_T}\right)}$$

$$C_{q,N} = \frac{\kappa\, C_{d,N}^{1/2}}{\ln\left(\frac{z_{ref}}{z_Q}\right)}$$

From these expressions, It can be deduced that increasing the coefficient of drag affects the coefficient of heat and humidity. Furthermore, from the bulk formulas, it can also be realized that this change in coefficient terms affects the sensible and latent heat fluxes, besides the effects of increasing wind speeds on the heat fluxes. Lastly, we would also like to point out that the parameterization used for the roughness length of temperature and humidity in the coupled and uncoupled experiments presented in our study are set equal to $0.95 \times 10^{-4}$ m, thus giving constant values for the denominator in the computation of coefficient of heat and humidity.

5. The examination of the sensitivity to spray focusses on heat fluxes, which have never been measured in hurricanes and therefore can't be verified. I suggest they could focus on the near-surface air temperature and humidity, which have been measured (e.g., we Zhang et al. 2017).

Ans: We would like to thank the reviewer for the suggestion. We would like to request more information on the study mentioned by the reviewer. Our search for the aforementioned study didn't lead to any useful results.

6. Comparisons of sensible and latent heat flux for Expts 2,3,4 suggest that spray has negligible effect on the thermodynamics. However, it is clear from fig 12b that much less spray is produced by the author's model compared to MOM80 in Andreas. I think it would be interesting to see a comparison of the total spray mass flux as a function of wind speed (this paper vs MOM80)

Ans: It is possible to infer it from the Figure 12b, where Figure 12b shows that the whitecap fraction is 25% of the whitecap fraction as opposed to the whitecap fraction obtained from MOM80. Because, in this study, we do not modify the SSGF, just use whitecap fraction from wave model instead of an empirical relation. As the whitecap fraction is merely used for scaling sea spray generation function for different wind speeds, therefore, in our view the effects of changing whitecap fraction should be applicable as a scaling parameter to the volume (or mass) flux of sea spray as well.

7. On a more editorial subject, I think the description of whitecap fraction and spray function parameters (sections 5.1.1 and 5.2.1) should be moved to section 2 since they are not part of the hurricane simulations.

Ans: Done. The two subsections are relocated to the section 2. Thank you for the suggestion.

**References:**

Andreas, E. L.: Spray-Mediated Enthalpy Flux to the Atmosphere and Salt Flux to the Ocean in High Winds, J. Atmospheric Sci., 40(3), 608–619, 2010.

Andreas, E. L., Persson, P. O. G. and Hare, J. E.: A Bulk Turbulent Air-Sea Flux Algorithm for High-Wind, Spray Conditions, J. Phys. Oceanogr., 38, 1581–1596, 2008.

Anguelova, M. D. and Hwang, P. H.: Using Energy Dissipation Rate to Obtain Active Whitecap Fraction, J. Phys. Oceanogr., 46(2), 461–481, 2016.

Bao, J. W., Fairall, C. W., Michelson, S. A. and Bianco, L.: Parameterizations of sea-spray impact on the air-sea momentum and heat fluxes, Mon. Weather Rev., 139(12), 3781–3797, 2011.

DeCosmo, J.: Air-sea exchange of momentum, heat, and water vapor over whitecap sea states, PhD Thesis, University of Washington, Seattle., 1991.

Fairall, C. W., Bradley, E. F., Rogers, D. P., Edson, J. B. and Young, G. S.: Bulk parametrization of air-sea fluxes for Tropical Ocean-Global Atmosphere Coupled-Ocean Atmosphere Response Experiment, J. Geophys. Res., 101, 3747–3764, 1996.

Fairall, C. W., Bradley, E. F., Hare, J. E., Grachev, A. A. and Edson, J. B.: Bulk Parameterization of Air–Sea Fluxes: Updates and Verification for the COARE Algorithm, J. Clim., 16(4), 571–591, doi:10.1175/1520-0442(2003)016<0571:BPOASF>2.0.CO;2, 2003.

Grant, A. L. M. and Hignett, P.: Aircraft observations of the surface energy balance in TOGA-COARE, Q. J. R. Meteorol. Soc., 124(545), 101–122, doi:10.1002/qj.49712454505, 1998.

Kilroy, G., Montgomery, M. T. and Smith, R. K.: The role of boundary-layer friction on tropical cyclogenesis and subsequent intensification, Q. J. R. Meteorol. Soc., 143(707), 2524--2536, doi:10.1002/qj.3104, 2017.

Kraan, G., Oost, W. A. and Janssen, P. A. E. M.: Wave energy dissipation by whitecaps, J. Atmospheric Ocean. Technol., 13(1), 262–267, 1996.

Liu, W. T., Katsaros, K. B. and Businger, J. A.: Bulk Parameterization of Air-Sea Exchanges of Heat and Water Vapor Including the Molecular Constraints at the Interface, J. Atmospheric Sci., 36(9), 1722–1735, doi:10.1175/1520-0469(1979)036<1722:BPOASE>2.0.CO;2, 1979.

Montgomery, M. T., Smith, R. K. and Nguyen, S. V.: Sensitivity of tropical-cylone models to the surface drag coefficient, Q. J. R. Meteorol. Soc., 136(653), 1945--1953, doi:10.1002/qj.702, 2010.

Mueller, J. A. and Veron, F.: Impact of sea spray on air–sea fluxes. Part II: Feedback effects, J. Phys. Oceanogr., 44(11), 2835–2853, 2014.

Persson, P. O. G., Hare, J. E., Fairall, C. W. and Otto, W. D.: Air-sea interaction processes in warm and cold sectors of extratropical cyclonic storms observed during FASTEX, Q. J. R. Meteorol. Soc., 131(607), 877–912, 2005.

---

## Author Comment (AC2) · 31 Jan 2018

Dr Troitskaya (Referee)

yuliya@hydro.appl.sci-nnov.ru

1. The model demonstrates strong sensitivity to the presence of waves. First, this fact needs explanation from the physical point of view

As described in Section 3.3, within the coupled model, the effects of ocean waves on the atmosphere model are included through the roughness length computed in the wave model. Due to the usage of wave model computed roughness length, the drag coefficient in atmosphere model includes the effects of the wave spectrum over a range of frequencies. This feedback between the atmosphere and wave model affects the dynamic structure of a TC by inducing non-linear interactions caused by the temporal and spatial variations in the roughness length (Katsafados et al., 2016), unlike the case of uncoupled model. In the case of coupled model, the roughness lengths are greater than those in uncoupled model, where this increase in roughness lengths increases both the sensible and latent heat flux in the atmosphere model (see Figure 13-15)

2. Second, investigation of sensitivity the wave model is strongly desirable, in particular sensitivity to the wind source term and wind dissipation term

As stated in the Section 1, this study focusses on the coupling between a wave model and an atmosphere model. Additionally, the effects of sea spray on the hurricane are also investigated. Therefore, the effects of different source terms used in the wave model are beyond the scope of this study. Furthermore, the wave model (DHI, 2012) used in this study provides only one formulation for the different source terms namely wind input and wave dissipation due to friction and whitecapping. We acknowledge that it is desirable to investigate the sensitivity of model results to the different formulations of source terms, however, due to the aforementioned reasons, it can't be carried out in the model setup used in this study.

3. Authors' estimations of the thermodynamic feedback coefficients ïA ͣα ͦ ïA ͨc' and ïA ᵍg˘ are in significant contract with another papers (eg., Andreas et al, 2008; Mueller, Veron, 2014). In this connection sensitivity study to the thermal feedback coefficients is needed.

The values of $\alpha, \beta$ and $\gamma$ are obtained following the procedure described in (Andreas and DeCosmo, 1999), however, it should be noted that in the present study, sea spray generation function depends on the whitecap fraction computed from wave model, whereas in Andreas et al. (2008), the sea spray generation function is based on Fairall et al. (1994) with empirically obtained whitecap fraction. The sea spray generation function in Fairall et al. (1994) uses Miller (1988) for bubble droplet source function and Monahan and Muircheartaigh (1980) for spume droplet source function. It has been shown within the paper that our model produces 25% sea spray droplets as compared to some other studies. Additionally, previous studies have relied on parametrisations for wave heights or wave spectrum, whereas our study uses the output from the wave model instead.

4. Sensitivity study to spray generation function is also desirable.

Another valid point. As described in the study, if we use a different SSGF, it has to be a function which can account for wave model output, which is why, we used the sea spray generation function from Miller (1998). The sensitivity study of SSGF has been left for the future studies, as this study was intended to develop a coupled atmosphere-wave model and investigate the effects of sea spray fluxes on tropical cyclone. Furthermore, this study was designed to highlight various steps necessary to couple sea spray model within a atmosphere model (coupled or uncoupled).

5. Explanation of the effect of spray on momentum exchange is to brief. An equation for horizontal velocity of spray droplet before falling back in ocean in Eq.(13) is needed. Besides, the starting lines of paragraph 2.2.3 consists of qualitative discussion of the effect of spray on momentum exchange in atmospheric boundary layer, but no quantitative description is given. It definitely should be explained in some details.

The horizontal velocity of the spray droplet $u_{sp}$ was given by

$$u_{sp} = \left(\frac{u_*}{\kappa}\right) \times \log\left(\frac{z_{sp}}{z_0}\right)$$

where height of spray droplet $z_{sp} = 0.63 H_s$, where significant wave height $H_s$ was obtained from the wave model, while friction velocity was used after adjusting for neutral stratification. The effects of spray on momentum exchange are shown in Figure 1 and 2, where Figure 1 shows the spatial variation on momentum

[Figure]

**Figure 1**. Plan views of surface momentum flux for (a) Expt. 1, (b) Expt. 2, (c) Expt. 3 and (d) Expt. 4 at 0000 UTC July 3, 2014.

flux in storm centric coordinates for the four numerical experiments whereas, Figure 2 shows the azimuthally averaged values of momentum flux to illustrate the radial variation of momentum fluxes.

[Figure]

**Figure 2.** Radial distribution of azimuthal averaged total momentum flux at 0000 UTC July 3, 2014

From Figure 2, it is clear that the coupling of wave model doubles the momentum flux as compared to the uncoupled atmosphere model. Furthermore, it is noticed that the highest momentum flux is obtained in the case when both spray mediated heat and momentum flux are applied to the coupled atmosphere-wave model.

6. Possibly some related references should be added
Ans: Done. Thank you for the suggested references.

**Bibliography:**

Andreas, E. L. and DeCosmo, J.: Sea spray production and influence on air-sea heat and moisture fluxes over the open ocean, in Air-Sea Exchange: Physics, Chemistry and Dynamics, pp. 327–362, Springer., 1999.

Andreas, E. L., Persson, P. O. G. and Hare, J. E.: A Bulk Turbulent Air-Sea Flux Algorithm for High-Wind, Spray Conditions, J. Phys. Oceanogr., 38, 1581–1596, 2008.

DHI: DHI MIKE 21. Spectral Wave module. Scientific Documentation, edited by MIKEbyDHI, DHI, Hørsholm, Denmark., 2012.

Fairall, C. W., Kepert, J. D. and Holland, G. J.: The Effect of Sea Spray on Surface Energy Transports over the Ocean, Glob. Atmosphere Ocean Syst., 2, 121–142, 1994.

Katsafados, P., Papadopoulos, A., Korres, G. and Varlas, G.: A fully coupled atmosphere–ocean wave modeling system for the Mediterranean Sea: interactions and sensitivity to the resolved scales and mechanisms, Geosci. Model Dev., 9(1), 161–173, doi:10.5194/gmd-9-161-2016, 2016.

Monahan, E. C. and Muircheartaigh, I.: Optimal power-law description of oceanic whitecap coverage dependence on wind speed, J. Phys. Oceanogr., 10(12), 2094–2099, 1980.